# *Enterococcus faecium* secreted antigen A generates muropeptides to enhance host immunity and limit bacterial pathogenesis

Byungchul Kim[1], Yen-Chih Wang[1], Charles W Hespen[1], Juliel Espinosa[1], Jeanne Salje[2], Kavita J Rangan[1], Deena A Oren[3], Jin Young Kang[4], Virginia A Pedicord[1,5†], Howard C Hang[1]*

[1]Laboratory of Chemical Biology and Microbial Pathogenesis, The Rockefeller University, New York, United States; [2]Centre for Tropical Medicine and Global Health, Nuffield Department of Medicine, University of Oxford, Oxford, United Kingdom; [3]Structural Biology Resource Center, The Rockefeller University, New York, United States; [4]Laboratory of Molecular Biophysics, The Rockefeller University, New York, United States; [5]Cambridge Institute of Therapeutic Immunology & Infectious Disease, University of Cambridge, Cambridge, United Kingdom

**\*For correspondence:**
hhang@rockefeller.edu

**Present address:** [†]Cambridge Institute of Therapeutic Immunology & Infectious Disease (CITIID), University of Cambridge, Cambridge, United Kingdom

**Abstract** We discovered that *Enterococcus faecium* (*E. faecium*), a ubiquitous commensal bacterium, and its secreted peptidoglycan hydrolase (SagA) were sufficient to enhance intestinal barrier function and pathogen tolerance, but the precise biochemical mechanism was unknown. Here we show *E. faecium* has unique peptidoglycan composition and remodeling activity through SagA, which generates smaller muropeptides that more effectively activates nucleotide-binding oligomerization domain-containing protein 2 (NOD2) in mammalian cells. Our structural and biochemical studies show that SagA is a NlpC/p60-endopeptidase that preferentially hydrolyzes crosslinked Lys-type peptidoglycan fragments. SagA secretion and NlpC/p60-endopeptidase activity was required for enhancing probiotic bacteria activity against *Clostridium difficile* pathogenesis *in vivo*. Our results demonstrate that the peptidoglycan composition and hydrolase activity of specific microbiota species can activate host immune pathways and enhance tolerance to pathogens.
DOI: https://doi.org/10.7554/eLife.45343.001

## Introduction

The microbiota provides an important barrier to enteric infections and encodes microbe-associated molecular patterns as well as secondary metabolites, which can prime host immunity or attenuate pathogen fitness (*Buffie and Pamer, 2013*; *Milshteyn et al., 2018*). For example, polysaccharide A (PSA) from *Bacteroides fragilis* has been reported to activate Toll-like receptor two on FOXP3[+] regulatory T cells, promote immunologic tolerance and enhance colonization of commensal bacteria (*Round et al., 2011*). Alternatively, butyrate production by *Clostridial* strains (clusters XIVa and IV) can attenuate inflammation by inducing peripheral regulatory T cells (*Arpaia et al., 2013*; *Furusawa et al., 2013*; *Smith et al., 2013*) and suppress reactive metabolites to prevent the expansion of enteric pathogens (*Byndloss et al., 2017*). In addition, commensal bacteria such as *Clostridium scindens* can generate secondary bile acids that mediate colonization resistance toward pathogens such as *Clostridium difficile* (*Buffie et al., 2015*). Other specific commensal strains such as *Enterococcus faecalis* (*E. faecalis*) contain plasmids such as pPD1 that produce bacteriocins, which can kill vancomycin-resistant enterococci and influence niche competition in the gut

(*Kommineni et al., 2015*). While these studies have begun to reveal the functions of some specific commensal bacteria species, the mechanisms of action and specific protective factors for other commensal bacteria that are correlated with pathogen susceptibility in animals are still unknown.

*E. faecium* is a ubiquitous bacterium that has been recovered from the microbiota of many animals as well as humans (*Van Tyne and Gilmore, 2014*) and can even colonize the roundworm *Caenorhabditis elegans* (*C. elegans*) (*Garsin et al., 2001*). While pathogenic strains of *E. faecium* and *E. faecalis* are notable due to their acquisition of drug-resistance and nosocomial infections (*Arias and Murray, 2012*), commensal strains of *E. faecium* have been reported to protect animals from enteric pathogens and potentially improve host metabolism (*Zheng et al., 2016*). However, the specific protective factors and mechanisms of commensal *E. faecium* were unknown (*Gilmore et al., 2014*). To dissect the protective mechanisms of *E. faecium*, we utilized *C. elegans* as an animal model and discovered that diverse strains of *E. faecium* could protect worms against enteric pathogens (*Rangan et al., 2016*). *E. faecium* and its growth media effectively prevented *Salmonella* Typhimurium-induced pathogenesis in *C. elegans*. The protective activity of *E. faecium* growth media was protease sensitive, which led to the proteomic analysis of the supernatant and discovery of secreted antigen A (SagA) as one of the most abundant *E. faecium* secreted proteins. Remarkably, administration of recombinant SagA to *C. elegans* was sufficient to attenuate *S.* Typhimurium pathogenesis (*Rangan et al., 2016*). In addition, chromosomal insertion of *sagA* into non-protective *E. faecalis* (*E. faecalis-sagA*) enabled the expression and secretion of SagA, which also conferred protective activity against *S.* Typhimurium pathogenesis in *C. elegans* (*Rangan et al., 2016*). *E. faecium* and SagA did not affect *S.* Typhimurium colonization or replication *in vivo*, but rather required *tol-1* signaling in *C. elegans*, suggesting the activation of host innate immunity was the key mechanism for enhanced tolerance against enteric pathogens (*Rangan et al., 2016*).

To evaluate whether *E. faecium* and SagA function through similar mechanisms in mammals, we evaluated germ-free and antibiotic-treated mouse models of infections. In these studies, *E. faecium* colonization, but not *E. faecalis,* improved intestinal barrier morphology, permeability, gene expression (Muc2, cryptdin2 and RegIIIγ) and prevented *S.* Typhimurium pathogenesis (*Pedicord et al., 2016*). *E. faecium* protection against *S.* Typhimurium did not involve adaptive immunity mechanisms, but required innate immune signaling factors such as MyD88 (Toll-like receptor adaptor) and NOD2 as well as RegIIIγ, a key effector of intestinal barrier function (*Pedicord et al., 2016*). *E. faecalis-sagA* also increased the expression of intestinal epithelial barrier function genes (Muc2, cryptdin2 and RegIIIγ) and enhanced bacterial segregation from intestinal epithelial cells in vivo similar to *E. faecium*. Moreover, SagA can be expressed in probiotic bacteria such as *Lactobacillus plantarum* and enhanced its protective activity against *S.* Typhimurium as well as *C. difficile* (*Pedicord et al., 2016*; *Rangan et al., 2016*). These studies demonstrated that *E. faecium* and SagA-expressing bacteria were sufficient to enhance intestinal barrier function and prevent infection by diverse Gram-negative and Gram-positive enteric pathogens, but the biochemical mechanism(s) by which innate immunity was activated was unknown.

Here, we demonstrated that *E. faecium* peptidoglycan contains smaller non-crosslinked muropeptides compared to non-protective strains of *E. faecalis*, which more effectively activated the intracellular peptidoglycan pattern recognition receptor NOD2. In addition, we determined the X-ray structure of the *E. faecium* SagA-NlpC/p60 catalytic domain and showed that it encodes L-Lys-type endopeptidase activity. The SagA-NlpC/p60 domain selectively cleaved purified crosslinked peptidoglycan fragments into smaller muropeptides, which were more active towards NOD2 in mammalian cells. Furthermore, we demonstrated that the secretion and peptidoglycan hydrolase activity of SagA was required to enhance probiotic bacteria activity against *C. difficile* infection *in vivo*. Our structural, biochemical, cellular and *in vivo* studies revealed how *E. faecium* and its secreted peptidoglycan hydrolase SagA activate innate immunity in mammals and provide a mechanistic basis by which commensal bacteria modulation of peptidoglycan composition confers host protection against enteric infections.

## Results

Since *E. faecium* protection against *S.* Typhimurium was abrogated in *Nod2-/-* mice *in vivo* and SagA contains predicted peptidoglycan hydrolase (*Pedicord et al., 2016*), we focused on whether *E. faecium* and SagA exhibit unique peptidoglycan composition and activity. As *sagA* was shown to be

essential for *E. faecium* growth (*Teng et al., 2003*), we focused on the characterization of *E. faecium* (Com15), *E. faecalis* (OG1RF), and *E. faecalis-sagA* (*Rangan et al., 2016*), in which the *sagA* gene was chromosomally inserted downstream of *mreD* and expressed under the endogenous *sagA* promoter. To directly compare SagA protein expression levels in these strains, we generated polyclonal sera against full length recombinant SagA (*Figure 1—figure supplement 1*) and confirmed that there were comparable levels of SagA expression in the cell pellets and secretion in the supernatants of *E. faecium* and *E. faecalis-sagA*, which was absent in *E. faecalis* (*Figure 1a*). Analysis of the bacterial growth rates showed that *E. faecium* divided most rapidly and reached optical density values of up to $OD_{600nm}$ = 2.5 after 4 hr (*Figure 1b*). During lag phase, there was no marked difference in *E. faecium*, *E. faecalis*, and *E. faecalis-sagA*. However, *E. faecalis-sagA* exhibited a decreased growth rate compared to *E. faecalis* after 3 hr, resulting in slower exponential growth (*Figure 1b*). Analysis of these bacterial strains by transmission electron microscopy showed that *E. faecium* exhibited unique cell wall morphology compared to *E. faecalis* (*Figure 1c*). Interestingly, *E. faecalis-sagA* exhibited a significant increase in bacterial cell width as well as the presence of minor deformations to the integrity of the cell wall structure (*Figure 1c,d*), suggesting SagA expression may regulate cell wall composition and integrity in *Enterococcus*.

To evaluate whether SagA expression affects bacterial cell wall composition, we isolated peptidoglycan from log-phase *E. faecium*, *E. faecalis*, and *E. faecalis-sagA* and analyzed the mutanolysin-digested peptidoglycan fragments by LC-MS/MS as previously described (*Kühner et al., 2014*). The major monomeric and crosslinked muropeptides we isolated from *E. faecium* and *E. faecalis* were similar to previous studies (*Billot-Klein et al., 1996*; *Emirian et al., 2009*; *Patti et al., 2008*), which showed that *E. faecium* generally contains non crosslinked muropeptides, including GlcNAc-Mur-NAc-L-Ala-D-isoGln (GlcNAc-MDP) compared to *E. faecalis* (*Figure 2a*). Interestingly, the analysis of peptidoglycan from *E. faecalis-sagA* revealed a decrease in muropeptide fragments (peak 2, 7, 9, 13, and 17) compared with *E. faecalis* (*Figure 2a*, *Figure 2—figure supplement 1a,b* and *Supplementary files 1* and *2*). To evaluate whether these changes in bacterial growth, cell morphology and peptidoglycan composition were associated with SagA expression and not other mutations generated during chromosomal *sagA* insertion, we performed whole genome sequencing on both *E. faecalis* and *E. faecalis-sagA*. This analysis confirmed the insertion of *sagA* downstream of *mreD*, as originally designed, and also revealed a total of 39 genes with synonymous and non-synonymous mutations ranging from 25 to 100 percent coverage (*Supplementary file 3*). However, none of the non-synonymous mutations greater than 50 percent coverage were in genes directly associated with bacterial cell wall synthesis or remodeling, suggesting SagA expression affected growth, morphology and peptidoglycan composition of *E. faecalis-sagA*.

We then tested whether *E. faecium*, *E. faecalis* and *E. faecalis-sagA* could activate peptidoglycan pattern recognition receptors. In mammals, NOD1 and NOD2 are the predominant sensors of peptidoglycan fragments and are reported to recognize gamma-D-glutamyl-meso-diaminopimelic acid (iE-DAP) and muramyl-dipeptide (MDP), respectively (*Caruso et al., 2014*; *Philpott et al., 2014*). Peptidoglycan activation of intracellular NOD1 and NOD2 triggers the expression of NF-κB-responsive genes associated with host immunity and inflammation (*Caruso et al., 2014*; *Philpott et al., 2014*). To evaluate the activation of these receptors, increasing amounts of *E. faecium*, *E. faecalis* and *E. faecalis-sagA* were added to NOD1 or NOD2-transfected HEK293T cells co-expressing NF-κB luciferase reporters. Compared to synthetic ligands iE-DAP and MDP, *E. faecium* with a multiplicity of infection (MOI = 1) activated NOD2, but not NOD1 (*Figure 2b*). Higher MOI of *E. faecium* resulted in the detachment of transfected HEK293T cells (data not shown). In contrast, *E. faecalis* and *E. faecalis-sagA* required $10^3$-fold more bacteria (MOI = $10^3$) to activate NOD2, but not NOD1 (*Figure 2b*). The selective activation of NOD2 is consistent with its agonist specificity for Lys-type muropeptides, which were found in *Enterococci* peptidoglycan (*Billot-Klein et al., 1996*; *Emirian et al., 2009*; *Patti et al., 2008*). While the chromosomal expression of SagA altered the peptidoglycan composition of *E. faecalis* (*Figure 2a*), these differences did not appear to be sufficient to enhance direct bacterial activation of NOD2 in HEK293T cells. The differences in NOD2 activation by *E. faecium* compared to *E. faecalis* and *E. faecalis-sagA* may be due to enhanced bacterial adhesion and/or internalization since more *E. faecium* was recovered from gentamicin protection assays (*Figure 2c*). These results showed *E. faecium* contained smaller muramyl peptides such as GlcNAc-MDP and may directly interact with mammalian cells to activate NOD2.

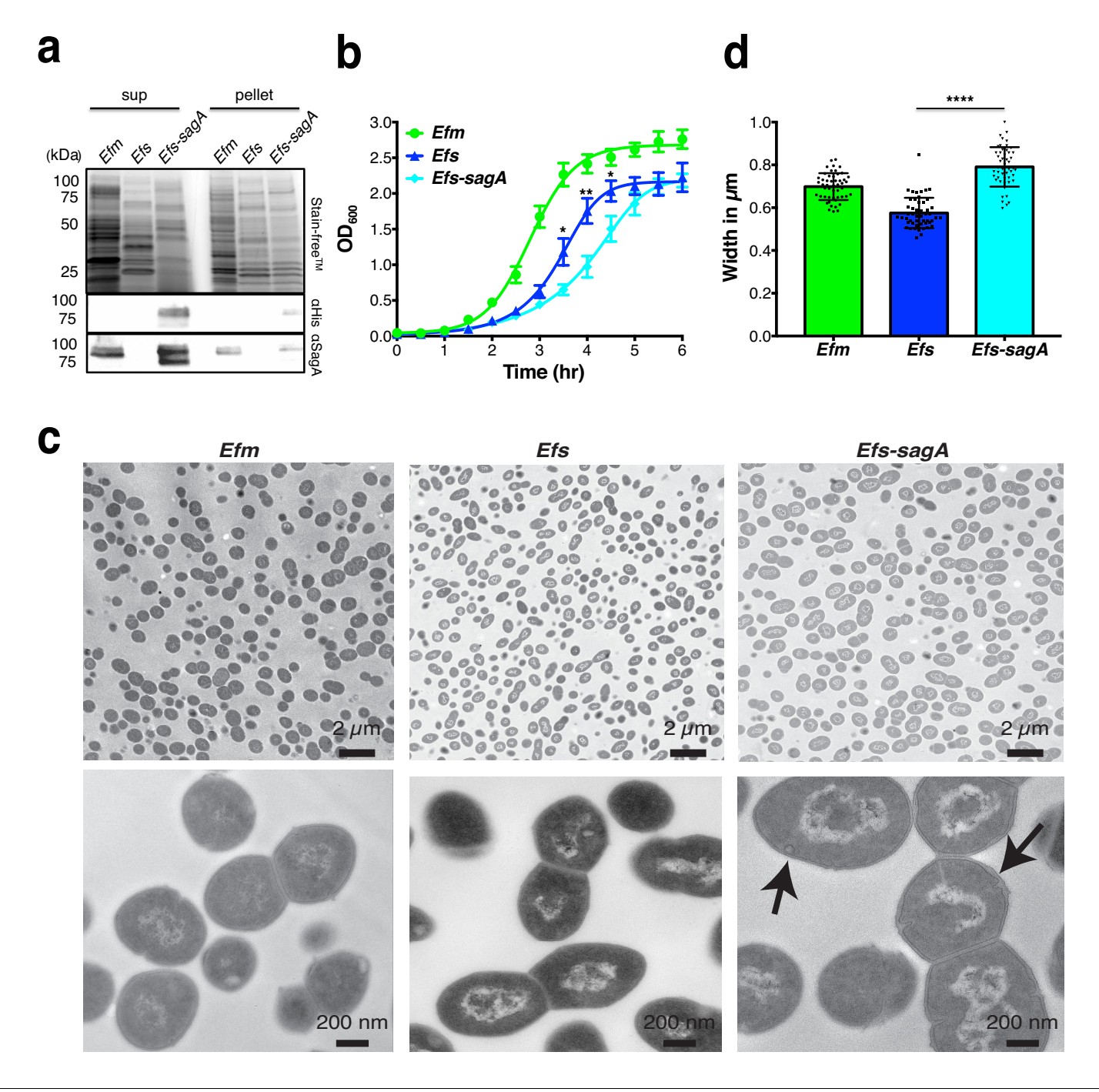

**Figure 1.** SagA expression alters *E.faecalis* growth and morphology. (a) Expression of SagA in *E. faecium* (*Efm*), *E. faecalis* (*Efs*), and *E. faecalis-sagA* (*Efs-sagA*) supernatant and cell pellet by anti-His$_6$ and anti-SagA blots. (b) Growth curve of *E. faecium*, *E. faecalis*, and *E. faecalis-sagA*. The error bars indicate standard deviation of triplicate measurements. Data was analyzed using a two-tailed *t*-test. *p≤0.05; **p≤0.01. (c) Electron microscopy of *Enterococci* strains. Cells for microscopy were grown in BHI medium at 37°C and collected in exponential growth phase. Top Panel: EM of *E. faecium*, *E. faecalis* and *E. faecalis-sagA* samples. Scale bars - 2 μm. Images of the *E. faecalis-sagA* revealed size differences from normal growth caused by the presence of the *sagA* expression. Bottom panel: Representative higher magnification images of each cell. Scale bars - 200 nm. The arrows indicate the morphological changes observed in the cell wall structure of *E. faecalis-sagA*. (d) 50 cells from *E. faecium*, *E. faecalis* and *E. faecalis-sagA* were randomly selected. Width was measured in each condition. Data was analyzed using an unpaired *t*-test with Welch's correction; *n* = 50 per group. *P* value < 0.0001.

DOI: https://doi.org/10.7554/eLife.45343.002

*Figure 1 continued on next page*

*Figure 1 continued*

The following figure supplement is available for figure 1:

**Figure supplement 1.** Western blot analysis of anti-SagA polyclonal sera.

DOI: https://doi.org/10.7554/eLife.45343.003

To further dissect the protective activity of *E. faecium* and SagA, we focused on the activity of purified SagA and soluble peptidoglycan fragments. SagA encodes an N-terminal signal sequence, predicted coiled-coil domain, Ser/Thr-rich linker region and C-terminal NlpC/p60-family peptidase domain (*Figure 3a*). Bioinformatic analysis suggested that SagA is unique to *E. faecium* (*Figure 3— figure supplement 1*), with no clear homologues with greater than 25% protein sequence identity in most sequenced strains of *E. faecalis* (*Neumann et al., 2019*; *Palmer et al., 2010*). Nonetheless, the SagA from *E. faecium* Com15 does share significant protein sequence similarity to other NlpC/p60 orthologs in other *E. faecium* strains (*Figure 3—figure supplement 2*). Indeed, we demonstrated that the NlpC/p60-family peptidase domain and conserved active site residues were required for SagA-mediated protection in *C. elegans* (*Rangan et al., 2016*). However, the precise peptidoglycan substrates of SagA were unknown.

To directly evaluate the peptidoglycan hydrolase activity of SagA, we expressed and purified full-length His-tagged SagA and truncated C-terminal His-tagged constructs along with C443A mutants (*Figure 3—figure supplement 3a,b*). These recombinant SagA constructs did not cleave purified intact *E. faecium* peptidoglycan, but incubation of the SagA-NlpC/p60 domain with mutanolysin-digested peptidoglycan yielded small muropeptides by in-gel fluorescence profiling (*Figure 3—fig-ure supplement 3c*). Intriguingly, the SagA-NlpC/p60 domain only cleaved mutanolysin-digested peptidoglycan from *E. faecium* and *E. faecalis*, but not *E. coli* (*Figure 3—figure supplement 3d*), which suggested that it may be specific for L-Lys-type peptidoglycan found in *Enterococcus* versus *m*DAP-type in *E. coli*. LC-MS analysis of the full-length SagA and SagA-NlpC/p60 reaction products revealed the generation of several new peptidoglycan fragments as enzymatic products such as tri-peptide (peak a), GlcNAc-MDP (peak b), GlcNAc-MurNAc-tri-di (peak c), GlcNAc-MurNAc-tetra-tri (peak d) and larger fragments (peak e) (*Figure 3—figure supplement 3e* and *Supplementary file 4*), which were absent when the full-length SagA-C443A and the SagA-NlpC/p60-C443A active site mutants were added. Full-length SagA expressed and purified from *E. coli* at similar protein concentrations was less active with mutanolysin-digested peptidoglycan than truncated SagA-NlpC/p60 domain (*Figure 3—figure supplement 3e*), suggesting the N-terminus may inhibit or modulate the activity of NlpC/p60 domain, akin to the coiled-coil domain of PcsB from *Streptococcus pneumonia* (*Bartual et al., 2014*).

To understand the molecular mechanism of *E. facium* SagA-NlpC/p60, we determined crystal structure of the *E. faecium* SagA-NlpC/p60 domain. Gel-filtration analysis revealed that SagA-NlpC/p60 is monomeric (*Figure 3b*) and amenable to crystallization (**Supplementary Results**). The structure of the SagA-NlpC/p60 domain was solved to 2.4 Å resolution by using X-ray crystallography. The data were indexed in space group P432 with one molecule per asymmetric unit (asu) and the initial phases was estimated by molecular replacement method by using *Staphylococcus aureus* CwlT (PDB 4FDY) as a search model. The electron density was well defined for the majority of the protein residues. Flexible segments of the first 26 residues, the $His_6$-tag, and some side chains were disordered and not included in the final model (*Figure 3c,d* PDB ID: 6B8C, *Supplementary file 5*). The SagA-NlpC/p60 domain contained the typical features of the cysteine peptidase fold (*Anantharaman and Aravind, 2003*), a central six-stranded, antiparallel β-sheet flanked by three surrounding α-helices (*Figure 3c*). The overall dimensions of the structure were about 32.5 × 30×23 Å (*Figure 3d*) and shared significant structural homology to other NlpC/p60 hydrolase domains. The root-mean-square deviations between superimposed Cα atoms was a range of 1.6 ~ 2.0 Å when comparing with NlpC/p60 hydrolase domains of other structures (*Supplementary file 6* and *Fig-ure 3—figure supplement 4*). The conserved catalytic triad (C443, H494 and H506) of the SagA-NlpC/p60 domain was located in a putative substrate-binding groove between the subdomains in the α2 and β3–β4 (*Figure 3c,d*), where electron density for these key amino acid residues were well observed (*Figure 3e*). An electrostatic potential analysis of the SagA-NlpC/p60 domain showed that the putative substrate-binding groove was mostly composed of negatively charged amino acid

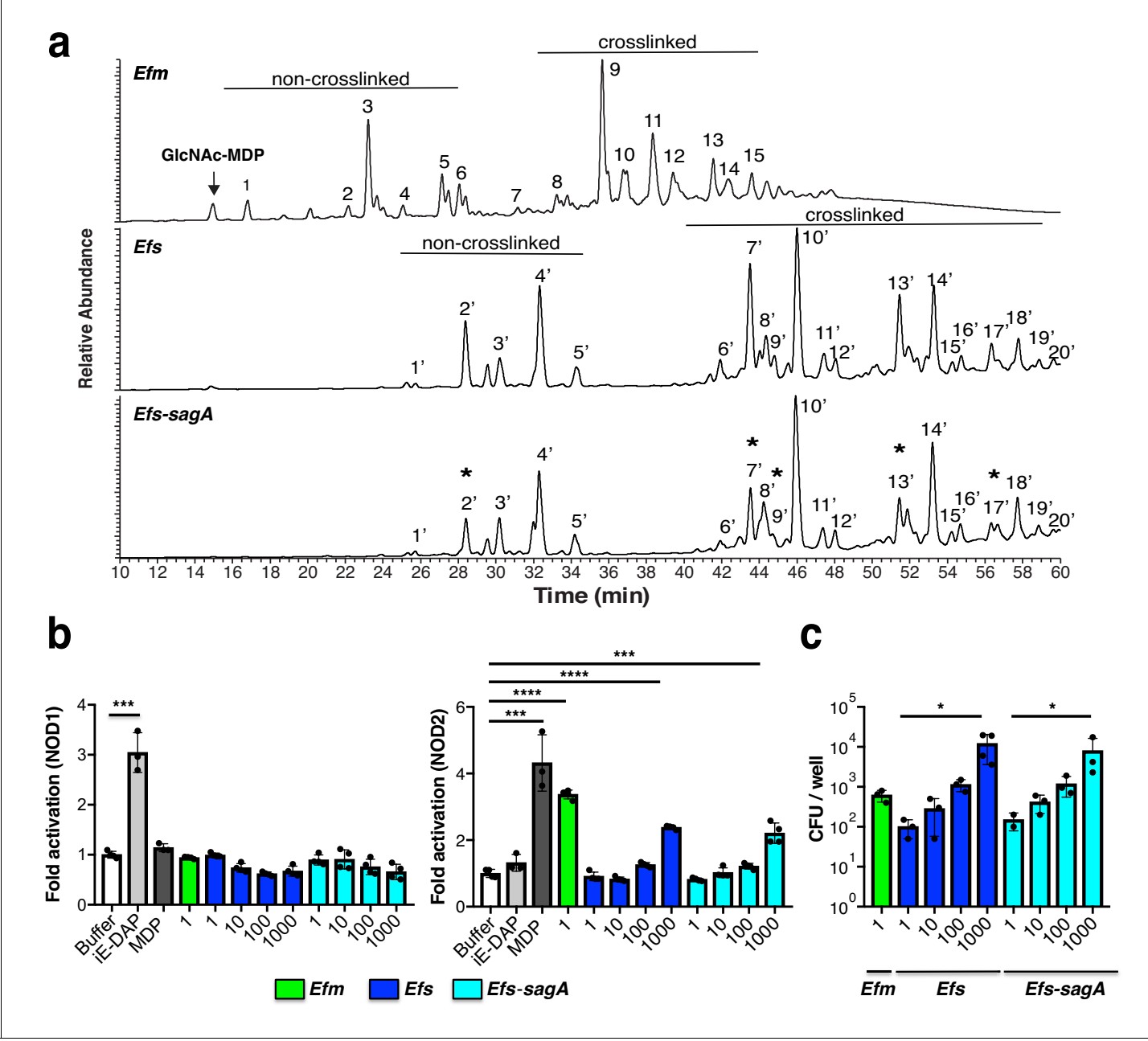

**Figure 2.** Analysis *Enterococci* peptidoglycan composition and activation of intracellular peptidoglycan pattern recognition receptors. (**a**) LC-MS analysis of peptidoglycan isolated from *E. faecium*, *E. faecalis*, and *E. faecalis-sagA* digested by mutanolysin. Numbers correspond to each muropeptide from *E. faecium*, *E. faecalis*, and *E. faecalis-sagA*. Muropeptides that are significantly decreased in *E. faecalis*-sagA compared to *E. faecalis* are marked with asterisks. Arrow indicates endogenous G-MDP peak from isolated *E. faecium* peptidoglycan. (**b**) Analysis of *E. faecium* (MOI = 1), *E. faecalis*, and *E. faecalis*-sagA (MOI = 1, 10, 100, 1000) activation of NOD1- and NOD2-expressing NF-κB reporter HEK293T cells with iE-DAP (50 μM, NOD1 ligand, light grey), MurNAc-L-Ala-D-isoGln (5 μM, MDP, NOD2 ligand, dark grey). (**c**) *E. faecium* (MOI = 1), *E. faecalis*, and *E. faecalis-sagA* (MOI = 1, 10, 100, 1000) internalization in HEK293T cells using gentamycin protection/CFU assay. For b and c, data are presented as means ± s.d.; *n* = 3 per group. Data was analyzed using a two-tailed *t*-test. *p≤0.05; **p≤0.01; ***p≤0.005; ****p≤0.001.
DOI: https://doi.org/10.7554/eLife.45343.004

The following figure supplement is available for figure 2:

**Figure supplement 1.** LC-MS analysis of mutanolysin-digested peptidoglycan fragments from *E. faecium*, *E. faecalis*, and *E. faecalis-sagA*.
DOI: https://doi.org/10.7554/eLife.45343.005

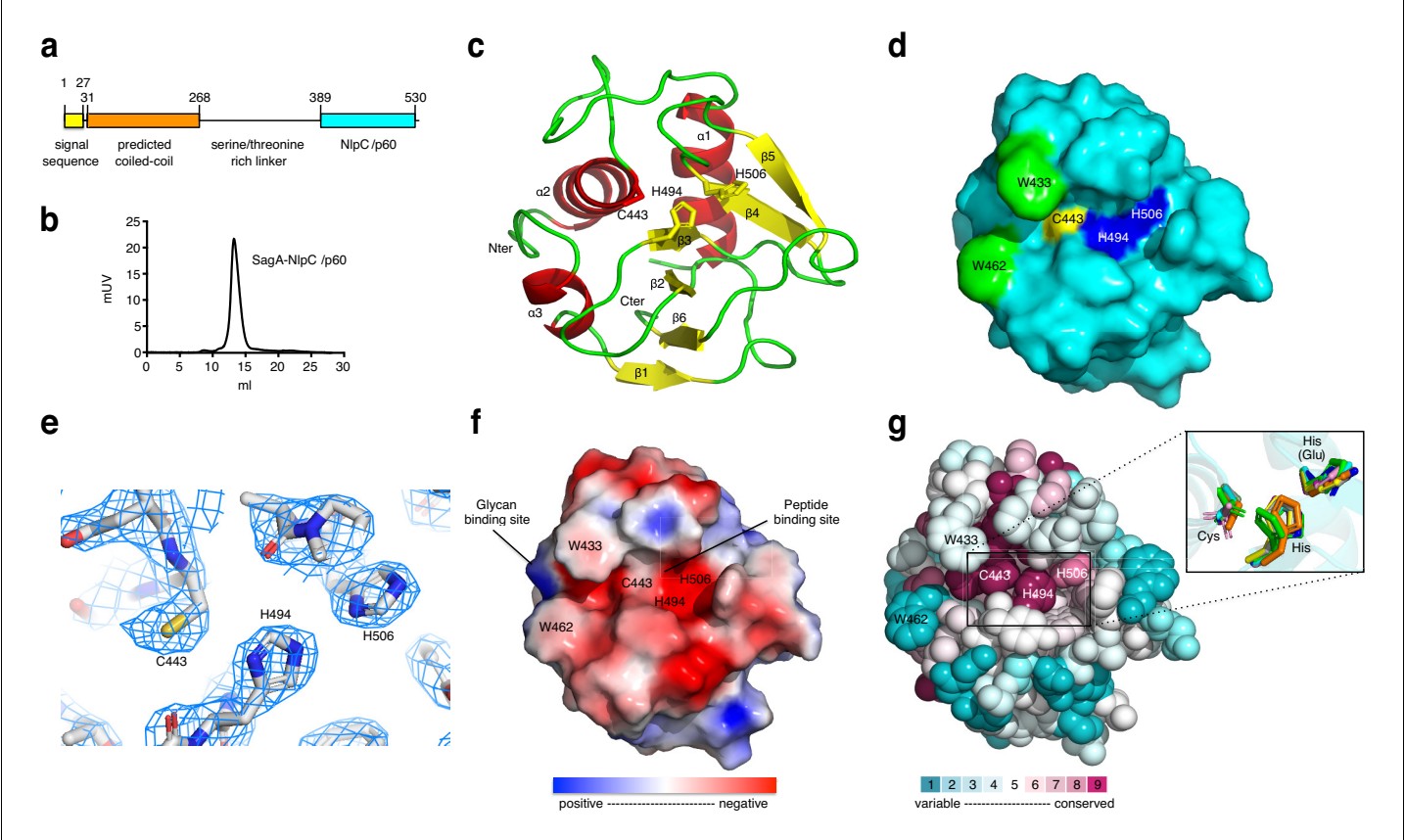

**Figure 3.** X-ray structure analysis of SagA-NlpC/p60 domain. (**a**) Schematic of SagA protein architecture, as predicted by Signal P, Jpred and BLASTP analysis. (**b**) Gel-filtration data of purified SagA-NlpC/p60. (**c**) Ribbon diagram of the C-terminal domain of SagA with secondary-structure elements labeled. (**d**) Surface representation of the SagA-NlpC/p60 (PDB entry: 6B8C). Catalytic triad of Cys443 (yellow), His494 (blue), His506 (blue), Trp433 (green), and Trp462 (green) are highlighted. (**e**) The $2F_o$-$F_c$ electron density (contoured at 1.5σ) of catalytic triad (Cys443, His494, and His506) is shown as a light blue mesh. (**f**) Electrostatic potentials of SagA-NlpC/p60. The color is scaled from −2 to 2 kT/e (blue, positive electrostatic potential; red, negative electrostatic potential) using PyMOL with the APBS tool. The surface shows potential glycan-binding site and substrate-binding groove, which can be proposed to bind to peptide from peptidoglycan fragment. (**g**) The conserved surface representation of SagA-NlpC/p60 is color coded according to amino acid conservation based on comparison to other homologs with sequence identities of 35% to 95% compared to SagA-NlpC/p60. Inlet box: Superimposition of the catalytic triad (Cys-His-His(Glu)) of SagA-NlpC/p60 with other structurally characterized peptidoglycan hydrolases. Color-coded are SagA-NlpC/p60 from *Enterococcus faecium* (cyan, PDB entry: 6B8C) with YkfC from *Bacillus cereus* (green, PDB entry: 3H41), NpPCP from *Nostoc punctiforme* (yellow, PDB entry: 2EVR), AvPCP from *Anabaena variabilis* (gray, PDB entry: 2HBW), CwlT from *Staphylococcus aureus* (pink, PDB entry: 4FDY), Spr from *Escherichia coli* (magenta, PDB entry: 2K1G), RipA from *Mycobacterium tuberculosis* (blue, PDB entry: 3NE0), and LysM from *Thermus thermophilus* (orange, PDB entry: 4XCM).
DOI: https://doi.org/10.7554/eLife.45343.006

The following figure supplements are available for figure 3:

**Figure supplement 1.** Multiple sequence alignment of C-terminal domains from different *Enterococcus faecalis* (*Efs*) strains.
DOI: https://doi.org/10.7554/eLife.45343.007
**Figure supplement 2.** Multiple sequence alignment of NlpC/p60 domains from different *Enterococcus faecium* (*Efm*) strains.
DOI: https://doi.org/10.7554/eLife.45343.008
**Figure supplement 3.** Characterization of peptidoglycan hydrolase activity of SagA constructs.
DOI: https://doi.org/10.7554/eLife.45343.009
**Figure supplement 4.** Alignment of SagA-NlpC/p60 domain with other homologs.
DOI: https://doi.org/10.7554/eLife.45343.010

residues (*Figure 3f*). Homology-based structural search using ConSurf server suggested that the residues with the highest percentage of conservation were in the active site in the putative substrate binding pocket. Structural alignments with other homologs also showed similar conformations of the catalytic triad in the active site to that of the YkfC from *Bacillus cereus* (*Xu et al., 2010*), NpPCP

from *Nostoc punctiforme* (*Xu et al., 2009*), AvPCP from *Anabaena variabilis* (*Xu et al., 2009*), CwlT from *Staphylococcus aureus* (*Xu et al., 2014a*), Spr from *Escherichia coli* (*Aramini et al., 2008*), RipA from *Mycobacterium tuberculosis* (*Ruggiero et al., 2010*), and LysM from *Thermus thermophilus* (*Wong et al., 2015*) active sites (*Figure 3g*).

To explore how the SagA-NlpC/p60 domain may interact with peptidoglycan substrates, we compared our structure to other NlpC/p60 hydrolases and performed potential substrate docking studies. While no structure of a NlpC/p60 hydrolase bound to peptidoglycan substrates has been determined, previous docking studies of CwlT suggested electropositive regions of these enzymes may be involved in substrate binding (*Xu et al., 2014a*). Indeed, the overlay of our SagA-NlpC/p60 structure with CwlT suggests potential conserved regions of substrate binding (*Figure 4a* and *Figure 3—figure supplement 4*). We then performed docking studies of solved structure with peptidoglycan fragments using Grid-based Ligand Docking with Energetics (GLIDE) module of the Schrödinger (*Friesner et al., 2006*). Amongst the peptidoglycan fragments we explored, GlcNAc-MurNAc-L-Ala-D-isoGln-L-Lys-D-Ala (*Figure 4b,c*) and the corresponding tetrapeptide alone (*Figure 4—figure supplement 1*) afforded a set of ligand-bound poses in the putative substrate-binding groove with high-scoring calculated binding free energies from 1 to 50 (*Supplementary file 7*). Manual inspection of the top-posed GlcNAc-MurNAc-tetrapeptide docked into the SagA-NlpC/p60 domain showed the predicted active site cysteine (C443) and histidine residues (H494 and H506) were well positioned near the potential site of peptide bond cleavage (*Figure 4c*, pose one from *Supplementary file 7*) and suggested peptidoglycan fragments may extend across the putative substrate-binding groove to interact with electropositive amino acid residues on the potential glycan and peptide binding sites at the surface (*Figure 3f*). Interestingly, the SagA-NlpC/p60 ligand-docked models also suggested that two tryptophans (W433 and W462) may serve as clamps for binding potential peptidoglycan substrates (*Figure 4b,c*). Indeed, single and double Ala mutants of W433 and W462 significantly abrogated hydrolytic activity of SagA-NlpC/p60 domain (*Figure 4—figure supplement 2*). These results demonstrate C443, W433 and W462 are crucial for the SagA-NlpC/p60 peptidoglycan hydrolase activity and encompass the enzyme active site as well as putative substrate-binding groove.

To identify the precise substrate of SagA-NlpC/p60, we further isolated muropeptides (disaccharide-tripeptide, disaccharide-tetrapeptide, disaccharide-pentapeptide, and crosslinked disaccharide-tripeptide-disaccharide-tetrapeptide) from *E. faecium* peptidoglycan. Disaccharide-tripeptide, disaccharide-tetrapeptide and disaccharide-pentapeptide were not cleaved or only showed modest production of GlcNAc-MDP (*Figure 4—figure supplement 3a,b,c*). However, incubation of SagA-NlpC/p60 with disaccharide-tripeptide crosslinked to disaccharide-tetrapeptide readily generated GlcNAc-MDP and the corresponding GlcNAc-MurNAc-L-Ala-D-isoGln-L-Lys-crosslinked-D-Ala-L-Lys heptapeptide product (GlcNAc-M7P) (*Figure 4d*) and *Figure 4—figure supplement 3d*) with a specific activity of approximately $67 \pm 7$ nmol/min/mg at 37°C (*Figure 4e*), which did not occur with the NlpC/p60-C443A mutant (*Figure 4—figure supplement 3d*). The identity of GlcNAc-MDP was further confirmed by LC-MS/MS analysis and its fragmentation into MDP (*Figure 4—figure supplement 4b,c*). These results demonstrated that SagA-NlpC/p60 encodes cysteine endopeptidase activity that prefers crosslinked peptidoglycan fragments and cleaves the peptide bond between D-iGln and L-Lys (*Figure 4f*).

We next evaluated the activity of the crosslinked muropeptide substrate and SagA-NlpC/p60-generated muropeptides, GlcNAc-MDP and GlcNAc-M7P, on NOD1 and NOD2 activation in mammalian cells. To control for endogenous activation, non-transfected parental HEK-Blue-Null2 cells were also used as a negative control (*Figure 5a*). None of SagA-generated ligands activated NOD1 (*Figure 5a*). We chemically synthesized GlcNAc-MDP (*Figure 5—figure supplement 1*) and directly compared its activity to SagA-generated GlcNAc-MDP and commercial MDP as a positive control. Using NOD1 and NOD2 HEK-Blue cells that exhibited better dose-response to ligands, we found that SagA-generated and synthetic GlcNAc-MDP activated NOD2 in a dose-dependent manner comparable to commercial MDP standard (*Figure 5b*). SagA-NlpC/p60-generated muropeptides, GlcNAc-MDP most effectively activated NOD2, and GlcNAc-M7P less effectively and both ligands were more active than the crosslinked peptidoglycan fragment (*Figure 5b*). These results demonstrated that SagA-generated muropeptides can directly activate NOD2 in mammalian cells, consistent with previous studies of MDP derivatives (*Dagil et al., 2016*; *Davis et al., 2011*; *Fujimoto et al., 2009*; *Girardin et al., 2003b*; *Wang et al., 2013*), which suggests that GlcNAc-

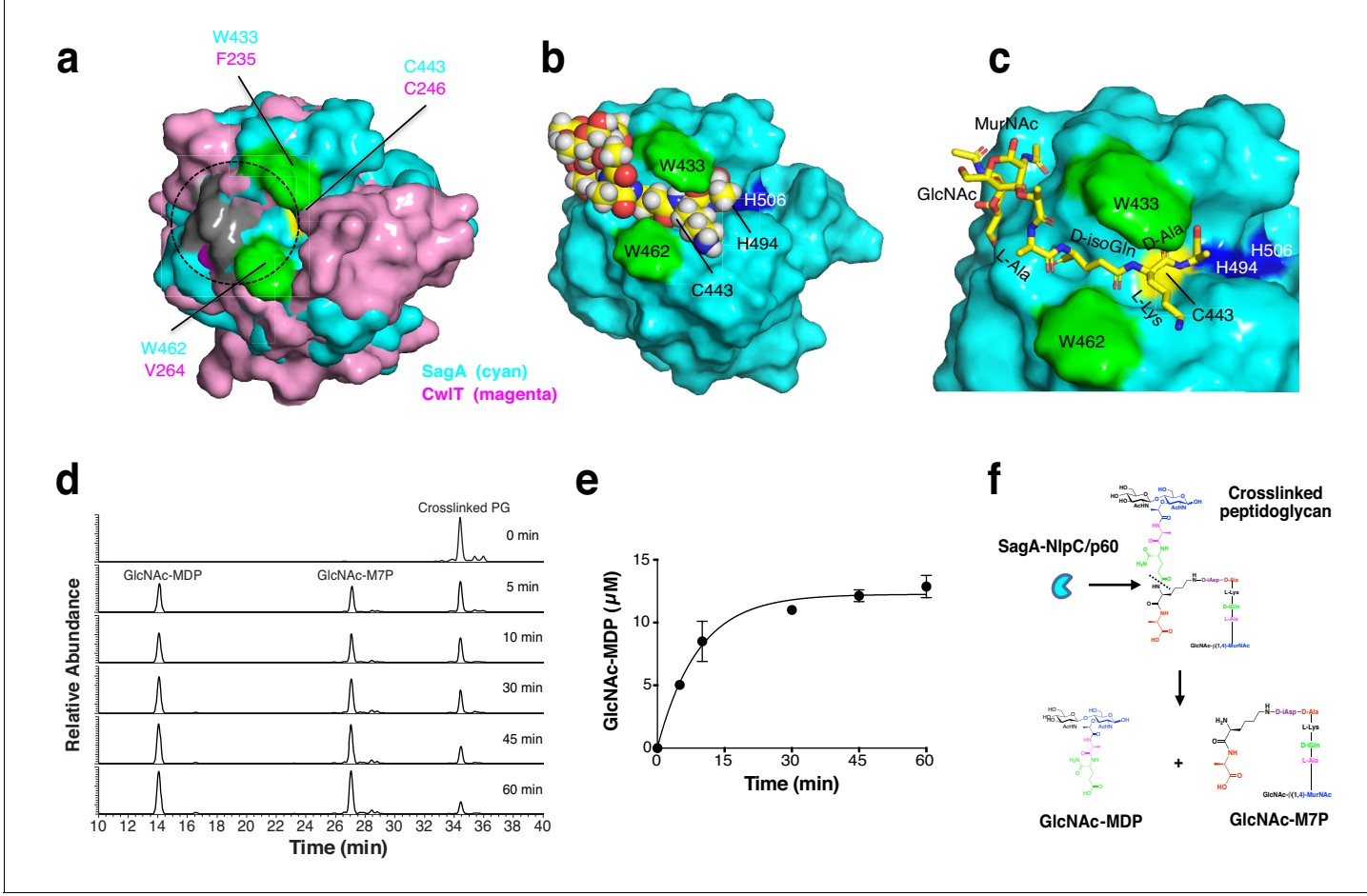

**Figure 4.** Model of peptidoglycan fragment bound to SagA and analysis of SagA-NlpC/p60 with purified peptidoglycan fragments. (a) Overlay of potential substrate binding site of SagA-NlpC/p60 (cyan) with CwlT (magenta). Positive charge residues are colored as light grey, aromatic (hydrophobic) residues as green, and catalytic cysteine residue as yellow. Predicted location of substrate-binding site is marked by circles. (b) Binding of peptidoglycan fragment (GlcNAc-MurNAc-tetrapeptide; L-Ala-D-isoGln-L-Lys-D-Ala) to SagA was modeled with space-filling representation using Glide (Schrödinger, LLC, New York, NY). Catalytic triad of Cys443 (yellow), His494 (blue), His506 (blue), Trp433 (green), and Trp462 (green) are highlighted. (c) Closer view of docked peptidoglycan fragment to SagA active site. (d) LC-MS analysis of reaction of isolated muropeptide purified from *E. faecium* by digesting with purified NlpC/p60 domain in time-dependent manner. Peak 1, purified crosslinked peptidoglycan (Disaccharide-tetrapeptide-disaccharide-tripeptide); 2, GlcNAc-MurNAc-L-Ala-D-isoGln-L-Lys-crosslinked-D-Ala-L-Lys heptapeptide product (GlcNAc-M7P); 3, GlcNAc-MurNAc-L-Ala-D-isoGln (GlcNAc-MDP). Products were confirmed by mass spectrometry. (e) Specific activity plot of SagA-NlpC/p60 with disaccharide-dipeptide. Specific activity was determined by quantification of product peak area using LC-MS. Data was obtained as the mean ± S.D. of the data from three independent experiments. (f) In vitro studies suggest SagA-NlpC/p60 domain cleaves crosslinked peptidoglycan fragments to generate GlcNAc-MDP and GlcNAc-M7P.

DOI: https://doi.org/10.7554/eLife.45343.011

The following figure supplements are available for figure 4:

**Figure supplement 1.** Model of peptidoglycan fragment by SagA.
DOI: https://doi.org/10.7554/eLife.45343.012
**Figure supplement 2.** Activity of SagA-NlpC/p60 mutants on mutanolysin-digested peptidoglycan from *E.faecalis*.
DOI: https://doi.org/10.7554/eLife.45343.013
**Figure supplement 3.** Analysis of SagA-NlpC/p60 with purified peptidoglycan fragments.
DOI: https://doi.org/10.7554/eLife.45343.014
**Figure supplement 4.** LC-MS analysis of SagA-NlpC/p60 activity on mutanolysin-digested peptidoglycan from *E. faecium*.
DOI: https://doi.org/10.7554/eLife.45343.015

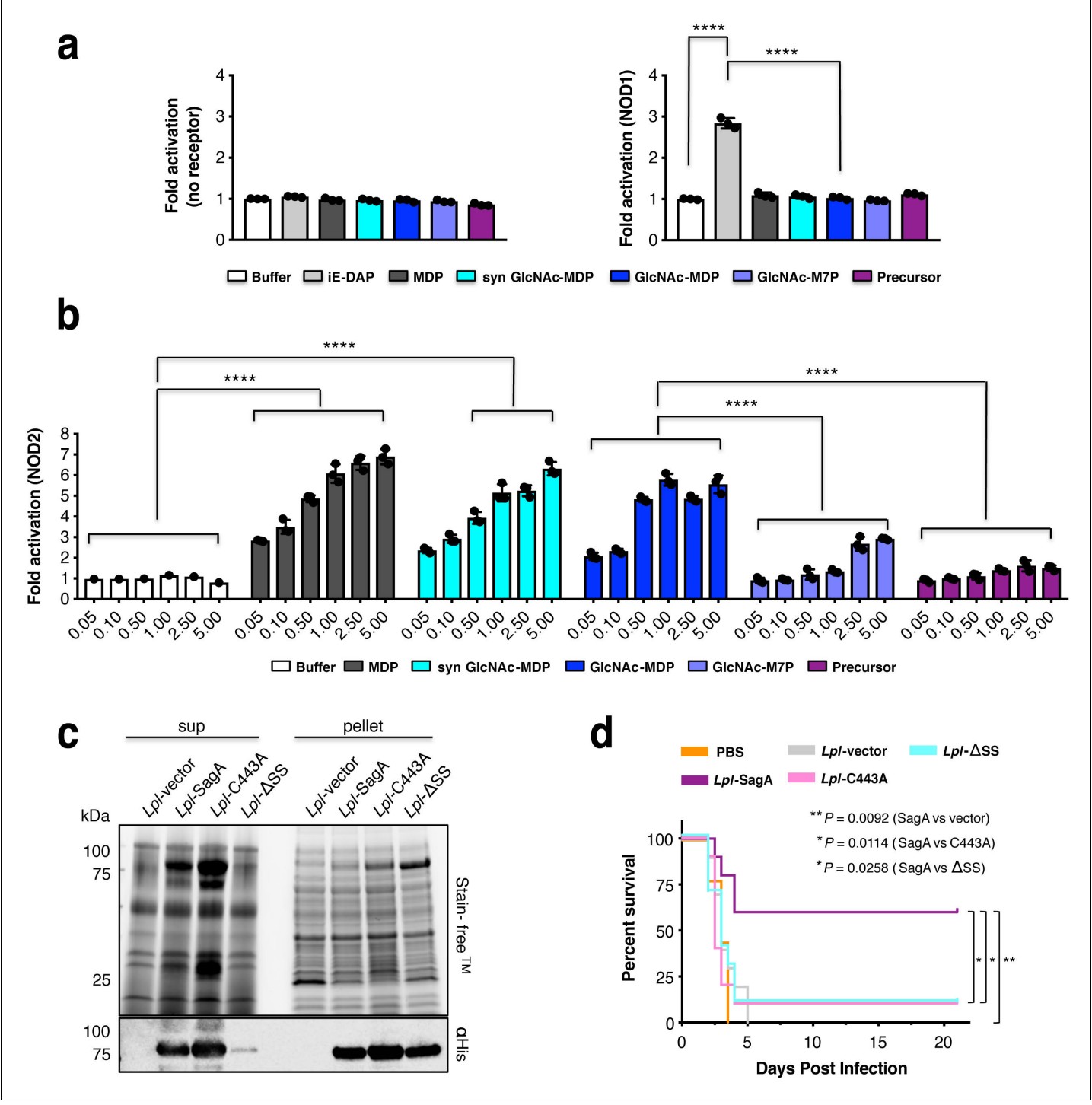

**Figure 5.** SagA activity generates small muropeptides which activate NOD2 signaling and can enhance *L. plantarum* probiotic activity against *C. difficile* infection in vivo. (**a**) HEK-Blue cells were treated with iE-DAP (50 μM, NOD1 ligand, light grey), MurNAc-L-Ala-D-isoGln (5 μM, MDP, NOD2 ligand, dark grey), synthetic GlcNAc-MurNAc-L-Ala-D-isoGln (5 μM, syn GlcNAc-MDP, cyan), crosslinked peptidoglycan fragment (disaccharide-tetrapeptide-disaccharide-tripeptide, purple) and products (GlcNAc-MurNAc-L-Ala-D-isoGln: GlcNAc-MDP or GlcNAc-MurNAc-L-Ala-D-isoGln-L-Lys-L-Lys-D-Ala: GlcNAc-M7P) at 5 μM for 12 hr. The measured firefly luciferase activity was divided by Renilla luciferase activity. The plotted values are relative ratios normalized to cells without ligand treatment, valued as 1. Data are shown as the mean ±SD from triplicate values. Data was analyzed using two-tailed *t*-test. *p≤0.05; **p≤0.01; ***p≤0.005; ****p≤0.001. (**b**) NOD2 activity in HEK-Blue cells with NF-κB activation after stimulation of cells with the MurNAc-L-Ala-D-isoGln (MDP), synthetic GlcNAc-MurNAc-L-Ala-D-isoGln, purified GlcNAc-MurNAc-L-Ala-D-isoGln, purified GlcNAc-MurNAc-L-Ala-D-isoGln-L-Lys-L-Lys-D-Ala, and purified crosslinked peptidoglycan fragment at different concentrations (0.05, 0.1, 0.5, 1, 2.5, 5 μM). Two-way

*Figure 5 continued on next page*

*Figure 5 continued*

ANOVA with a Sidak's posttest comparing buffer groups to MDP, syn GlcNAc-MDP, GlcNAc-MDP and GlcNAc-MDP to GlcNAc-M7P, crosslinked peptidoglycan fragment groups. Buffer shown as a negative control. *p≤0.05; **p≤0.01; ***p≤0.005; ****p≤0.001 for all analyses. (c) Expression of wild-type SagA and mutants in *L. plantarum* cell pellet and supernatant by anti-His$_6$ western blot. (d) Mice were given antibiotics (ampicillin, metronidazole, neomycin, vancomycin (AMNV) for 7 days and colonized with $10^8$ CFU of indicated bacteria 36 hr prior to oral infection with $10^6$ *C. difficile*. Pooled data from three independent experiments, n = 9–10 mice/group. Survival curve of *C. difficile* infected mice. Log-rank analysis, p-value shown comparing *L. plantarum* expressing SagA, compared to vector control, C443A, signal sequence mutant (ΔSS), respectively. *p≤0.05, **p≤0.01, ***p≤0.001 for all analyses.

DOI: https://doi.org/10.7554/eLife.45343.016

The following figure supplements are available for figure 5:

**Figure supplement 1.** Schematic of GlcNAc-MurNAc-L-Ala-D-isoGln (GlcNAc-MurNAc-dipeptide) synthesis.

DOI: https://doi.org/10.7554/eLife.45343.017

**Figure supplement 2.** Chemical compound 3.

DOI: https://doi.org/10.7554/eLife.45343.018

**Figure supplement 3.** Chemical compound 4.

DOI: https://doi.org/10.7554/eLife.45343.019

**Figure supplement 4.** Chemical compound 5.

DOI: https://doi.org/10.7554/eLife.45343.020

**Figure supplement 5.** Chemical compound 6.

DOI: https://doi.org/10.7554/eLife.45343.021

**Figure supplement 6.** Chemical compound 7.

DOI: https://doi.org/10.7554/eLife.45343.022

**Figure supplement 7.** Western blot analysis of *E. faecalis-sagA* active site mutant.

DOI: https://doi.org/10.7554/eLife.45343.023

**Figure supplement 8.** LC-MS analysis of mutanolysin-digested peptidoglycan fragments from *L. plantarum*.

DOI: https://doi.org/10.7554/eLife.45343.024

**Figure supplement 9.** Individual weight loss of antibiotic-treated mice against *C. difficile* infection in vivo.

DOI: https://doi.org/10.7554/eLife.45343.025

MDP derivatives are further processed by hydrolytic enzymes to generate MDP in mammalian cells. Importantly, these results indicated non-crosslinked muropeptides more effectively activate NOD2 in mammalian cells and provides a biochemical mechanism by which *E. faecium* and secreted SagA may enhance host immunity toward enteric pathogens.

To determine if SagA cleavage of peptidoglycan is important in vivo, we evaluated the protective activity of bacteria expressing wild-type or SagA mutants in mouse model of *C. difficile* infection. Since *sagA* is essential in *E. faecium* (*Teng et al., 2003*) and oral administration of recombinant SagA was not effective (data not shown), we focused on heterologous expression of SagA variants in *E. faecalis* and the probiotic bacteria *L. plantarum* (*Lpl*) (WCSF1). While chromosomal insertion of *sagA* into *E. faecalis* was functional (*Rangan et al., 2016*), insertion of a SagA active site mutant yielded low or undetectable levels of protein expression in cell lysate or supernatant (*Figure 5—figure supplement 2*). Overexpression of SagA variants on plasmids in *E. faecalis* were either toxic or did not yield adequate levels of protein expression (data not shown). However, plasmid-based expression of SagA-His$_6$ variants in *L. plantarum* yielded comparable levels of wild-type SagA, signal sequence mutant (ΔSS) as well as active site mutant (C443A) in cell lysates, with reduced levels of ΔSS mutant in the supernatant (*Figure 5c*). Heterologous expression of these SagA variants did not significantly alter the *L. plantarum* peptidoglycan composition (amidated *m*DAP-type) (*Goffin et al., 2005*) (*Figure 5—figure supplement 3*). However, colonization of antibiotic (Abx)-treated mice with these *L. plantarum* strains showed that wild-type SagA protected mice from *C. difficile* pathogenesis, while the SagA signal sequence (ΔSS) as well as active site (C443A) mutant were not protective and comparable to *L. plantarum*-vector and PBS controls (*Figure 5d* and *Figure 5—figure supplement 4*). These results demonstrate that SagA secretion and peptidoglycan endopeptidase activity are required for enhancing the activity of probiotic bacteria against enteric pathogens in vivo.

## Discussion

The microbiota and specific strains of commensal and probiotic bacteria have been suggested to activate peptidoglycan pattern recognition receptors in vivo (*Caruso et al., 2014*; *Philpott et al., 2014*). Moreover, deletion or loss-of-function mutations of NOD2 in particular are associated with gut microbiota dysbiosis, invasion of pathobionts and inflammatory bowel disease (IBD) (*Caruso et al., 2014*; *Philpott et al., 2014*), which suggests that NOD2 is important for sensing peptidoglycan in the gut and priming innate immune pathways to maintain intestinal barrier function and preventing microbiota- and pathogen-induced inflammation (*Al Nabhani et al., 2017*; *Caruso et al., 2014*; *Philpott et al., 2014*). Indeed, our previous studies showed that NOD2 was required for *E. faecium*-mediated protection against enteric pathogen infection (*Pedicord et al., 2016*). However, the mechanism by which *E. faecium* and its secreted peptidoglycan hydrolase SagA engaged NOD2 was unknown.

To dissect the biochemical mechanism(s) by which *E. faecium* and SagA enhances intestinal barrier function and protects against enteric pathogens, we evaluated *E. faecium* peptidoglycan composition, SagA structure, their associated muropeptide products, activation of peptidoglycan pattern recognition receptors in mammalian cells. The comparative analysis of the peptidoglycan composition and NOD2 activity between bacteria suggested that *E. faecium* more effectively activated NOD2 through non-crosslinked muropeptides such as GlcNAc-MDP that was not present in *E. faecalis and L. plantarum,* even with SagA expression (*Figure 2* and *Figure 5—figure supplement 3*). The results suggested *E. faecium* may be intrinsically more capable of interacting with mammalian cells and/or directly activate NOD2 even though both *Enterococci* species have Lys-type peptidoglycan. Nonetheless, SagA expression still enhances the protective activity of *E. faecalis* and *L. plantarum* against enteric pathogens in vivo (*Pedicord et al., 2016*; *Rangan et al., 2016*), suggesting secreted SagA may also prime or activate immune pathways.

To characterize the activity of secreted SagA, we expressed and purified recombinant SagA and investigated on the structure and enzymatic activity of C-terminal NlpC/p60 hydrolase domain. Although full-length SagA purified from *E. coli* was active, we focused on the C-terminal NlpC/p60 hydrolase domain since it was significantly more active *in vitro*. X-ray crystallography showed that the SagA-NlpC/p60 domain was indeed similar to other bacterial hydrolases and shared similar active site residues (*Figure 3*). Analysis of SagA-NlpC/p60 domain with purified peptidoglycan fragments demonstrated that it showed endopeptidase activity and preferentially cleaved L-Lys-type crosslinked peptidoglycan fragments (*Figure 4*), which required the conserved active site Cys443. In addition, W433 and W462 were also essential for the endopeptidase activity *in vitro* and may be involved in stabilization of the peptidoglycan substrate binding based on our SagA-NlpC/p60 structure and docking studies. These tryptophan residues or other hydrophobic aromatic amino acids are highly conserved amongst other SagA orthologs in *Enterococci* as well as other hydrolases in other bacteria (*Figure 3—figure supplements 1* and *2*). Beyond these highly conserved amino acid residues in the SagA-NlpC/p60 active site, the molecular basis for Lys- or DAP-type peptidoglycan specificity is still unclear from ours and other existing NlpC/p60 hydrolases structures (*Xu et al., 2009*; *Xu et al., 2010*; *Xu et al., 2014b*; *Xu et al., 2015*). Additional studies of these SagA-NlpC/p60 mutants with crosslinked muropeptides are therefore in progress to fully understand their roles in catalysis and substrate recognition. Further studies of full-length SagA are also needed to investigate its structure and activity as well as potential auto-inhibition by the N-terminal coil-coil domain observed in other NlpC/p60 hydrolases (*Bartual et al., 2014*). For example, the X-ray structure of full-length *Streptococcus pneumoniae*, PcsB, another class of peptidoglycan hydrolase that contains an N-terminal coil-coil domain, suggest this domain facilitate its dimerization and regulate activity as well as bacterial cell division (*Bartual et al., 2014*). The function of SagA and its orthologs in *Enterococci* peptidoglycan turnover and cell division remains to be determined, since *sagA* is essential in *E. faecium* (*Teng et al., 2003*) and requires new approaches for conditional inactivation for further studies.

Notably, these studies demonstrated that SagA-NlpC/p60 generated non-crosslinked muropeptides such as GlcNAc-MDP, which were also found in *E. faecium* peptidoglycan and activated NOD2 in mammalian cells more effectively than larger and crosslinked muropeptides (*Figure 5*). In addition to these *in vitro* biochemical studies and ex vivo activation of NOD2 in mammalian cells, we demonstrate that the secretion and endopeptidase activity of SagA is required for enhancing the protective

activity of *L. plantarum* against *C. difficile* infection *in vivo* (*Figure 5d*). Collectively, our results suggest that *E. faecium* and its secreted peptidoglycan hydrolase SagA generates small muropeptides that activate NOD2, which triggers downstream signaling pathways to enhance intestinal barrier function and host immunity (*Figure 6*). Although MDP were not active in *C. elegans*, which does not have a clear NOD2 ortholog, MurNAc and MurNAc-L-Ala were sufficient to attenuate bacterial pathogenesis and required the Tol-1 signaling (*Rangan et al., 2016*), suggesting the recognition of small bacteria-specific glycans may also be conserved in nematodes but utilize different receptors akin to insects (*Dziarski and Gupta, 2006*). In mammals, smaller muropeptide fragments have been reported to enhance NOD2 activation *ex vivo* (*Dagil et al., 2016*; *Davis et al., 2011*; *Fujimoto et al., 2009*; *Girardin et al., 2003a*; *Wang et al., 2013*) and function as more effective adjuvants for vaccination (*Hancock et al., 2012*; *Ogawa et al., 2011*; *Rubino et al., 2013*) or enhance monocyte activity *in vivo* (*Coulombe et al., 2012*; *Namba et al., 1996*). Since NOD2 is important for modulating immune signaling in many areas of host physiology and disease (*Caruso et al., 2014*; *Keestra-Gounder and Tsolis, 2017*; *Philpott et al., 2014*), further analysis of *Enterococci* and SagA orthologs will be important for understanding specific microbiota-host interactions and may afford new opportunities for therapeutic development.

## Materials and methods

### Growth curve

For comparison between bacterial growth in the medium, the stationary-phase bacterial suspension was diluted using fresh BHI medium to yield a similar bacterial concentration. 1 ml of this dilution was added to 100 ml of fresh 37°C pre-warmed BHI medium. The new suspension was incubated at 37°C, and the change in bacterial population in the medium with time (the growth curve) was measured at 0.5 hr intervals. Optical density measurements (at $OD_{600}$) were taken at each time point using a SpectraMax M2 spectrophotometer (Molecular Devices).

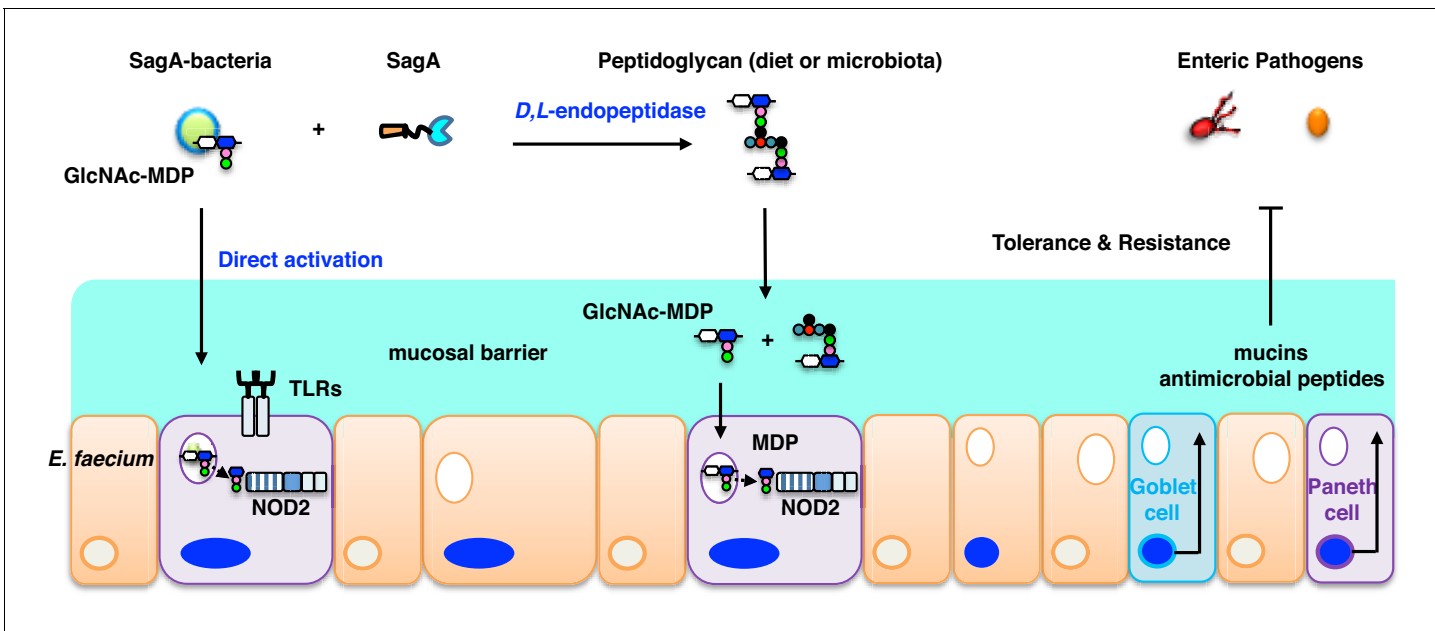

**Figure 6.** *E. faecium* and SagA generate small muropeptides that activate NOD2 signaling to inhibit enteric pathogens. Endogenous GlcNAc-MDP from *E. faecium* can stimulate NOD2 directly and secreted peptidoglycan hydrolase SagA from *E. faecium* can also generate small muropeptides that can activate NOD2. Activated NOD2 controls immunity and barrier function in the gut to protect the host against enteric pathogens.
DOI: https://doi.org/10.7554/eLife.45343.026

## Generation of anti-SagA polyclonal sera

The anti-SagA polyclonal sera were generated by Pocono Farm and Rabbit company. Briefly, rabbits were immunized by multiple intradermal injections with a total of 1 ml of purified full-length SagA (1 mg/ml) in saline emulsified with an equal volume of complete Freund's adjuvant. 100 μg of recombinant SagA emulsified in Freund's incomplete adjuvant was administered subcutaneously twice every 2 weeks. Animals were bled monthly and the sera stored at 4°C until required. Approximately 5 ml of blood was collected prior to each boost and the antibody titers in the sera were determined by enzyme-linked immunosorbent assay (ELISA). After the final boosts were administered, the rabbits were sacrificed and their blood collected. It is the antisera prepared from these final bleeds that were used to carry out the western-blot analysis.

## Western blot analysis of bacterial pellets and supernatant

Proteins were separated by SDS-PAGE on 4–20% Criterion TGX precast gels (Bio-Rad), then transferred to nitrocellulose membrane (0.2 μM, BioTrace NT Nitrocellulose Transfer Membranes, Pall Laboratory). HRP conjugated polyclonal anti-His$_6$ (abcam, ab1187) and polyclonal SagA serum as a primary antibody/HRP conjugated anti-Rabbit IgG (GE Healthcare, NA 934V) as a secondary antibody were used for His$_6$ and SagA blots respectively. Polyclonal SagA primary antibodies were used at a 1:50,000 dilution and secondary antibody at 1:10,000, unless otherwise stated. Membranes were blocked for 1 hr in 1% milk, incubated with primary antibody in 1% milk for 1 hr, washed 5 × with TBS-T (Tris-buffered saline, 0.1% Tween 20), incubated with secondary antibody, and washed with 4 × TBS T. Protein detection was performed with ECL detection reagent (GE Healthcare) on a Bio-Rad ChemiDoc MP Imaging System.

## Electron microscopy

Bacteria were inoculated from glycerol stocks onto BHI plates and incubated at 37°C overnight. Individual colonies were grown in 2 ml BHI media at 37°C and 220 rpm overnight. Bacterial cultures were diluted 1/10-1/50 in fresh BHI media and incubated at 37°C and 220 rpm for a further 3–5 hr until they reached mid-exponential phase (OD$_{600}$ ~0.4–0.7). 1 ml bacteria were combined with 1 ml 2x fixative (8% paraformaldehyde, 4% glutaraldehyde, 0.2 M sodium cacodylate, pH 7.4) and incubated at room temperature for 10–15 min. Samples were pelleted, supernatant removed, and pellets were resuspended in 1 x fixative (4% paraformaldehyde, 2% glutaraldehyde, 0.1 M sodium cacodylate, pH 7.4). Then pellets were post-fixed in 1% osmium tetroxide for 1 hr on ice. Fixation, ethanol dehydration, infiltration and polymerization were microwave processed using a Pelco Biowave (Ted Pella). The dehydration series was performed using 50%, 70%, 3 × 100% ethanol and 100% acetone. Bacteria were infiltrated with 1:1 LR white/acetone and two additional exchanges of LR White. Silver sections were picked up on formvar coated mesh grids, post-stained using 2 % UA and imaged on a JEOL 100CX with an AMT XR41 digital imaging system at 80kV.

## Overexpression and purification of SagA constructs

For *E. coli* BL21-RIL (DE3), 1L LB cultures were inoculated with overnight cultures with appropriate antibiotics, grown for 2 hr or until OD$_{600}$ ~0.5, induced with 1 mM isopropyl-*D*-thiogalactopyranoside (IPTG), then grown for an additional 2 hr. Cells were collected and resuspended in 20 ml of lysis buffer (20 mM Tris-HCl, pH 8.0, 150 mM NaCl, 0.1% SDS, 0.025 U/ml benzonase, and 1x protease inhibitor cocktail). After 15 min of sonication followed by centrifugation at 30,000 x *g* for 30 min, the supernatant containing the soluble target protein was collected and loaded onto a His60 Ni Superflow resin (Takara Bio) equilibrated with the binding buffer (PBS buffer) for a 1 hr incubation. The target protein was eluted with 300 mM imidazole. Semi-purified protein was dialyzed into PBS buffer at 4°C overnight using 10K MWCO Slide-A-Lyzer MINI dialysis devices (Thermo Fisher Scientific). Protein was further purified by loading onto a Superdex 75 Increase 10/300 GL column (GE Healthcare) pre-equilibrated with PBS. Fractions containing the target protein were combined and concentrated to 17 mg/ml for crystallization. Protein concentration was estimated by BCA assay (Pierce Protein Biology) and protein was stored at −80°C in PBS buffer and 10% glycerol.

For *E. coli* BL21-RIL (DE3) encoding full length SagA-His$_6$ constructs, 300 ml LB culture media was collected after 2 hr of IPTG-induced expression and secretion of the target protein. The culture supernatant containing the secreted target protein was vacuum filtered through 0.2 μm membrane

and then added to Jumbosep centrifugal devices with 10K MWCO membrane inserts (Pall Life Sciences) for fivefold concentration and buffer exchange to PBS. The protein concentrate was collected and loaded onto a His60 Ni Superflow resin pre-equilibrated with PBS for overnight incubation and subsequent affinity chromatography and dialysis as described above. The protein was further purified by loading onto a Superdex 200 Increase 10/300 GL column (GE Healthcare) pre-equilibrated with PBS. Fractions containing the target protein were combined and concentrated to 2 mg/ml for in vitro activity assays. Protein concentration estimation and protein storage were performed as described above.

## Peptidoglycan purification

Bacteria (*E. faecium* and *E. faecalis*) were grown in fresh BHI medium with shaking at 37°C to log-phase ($OD_{600}$ of 0.6). Peptidoglycan was extracted by resuspending the bacterial cell pellet in 0.25% SDS solution in 0.1 M Tris-HCl, pH 6.8 and boiling the suspension for 20 min at 100°C in a heating block as previously described (*Kühner et al., 2014*). The resulting insoluble cell wall preparation was washed with distilled water six times until free of SDS. The cell wall was purified by treatment with benzonase followed by trypsin digestion. Then, insoluble cell wall was recovered by centrifugation (16,000 x g, 10 min, 4°C), and washed once in distilled water. To obtain pure peptidoglycan, cell wall was then suspended in 1 M HCl and incubated for 4 hr at 37°C in a shaker to remove wall teichoic acid. The insoluble material was collected by centrifugation (16,000 x g, 10 min, 4°C) and washed with distilled water repeatedly until the pH was 5–6. The final peptidoglycan was lyophilized and stored at −20°C. For muropeptide analysis, purified peptidoglycan was digested with mutanolysin from *Streptomyces globisporus* (Sigma, 10 KU/ml of mutanolysin in $ddH_2O$) in 10 mM sodium phosphate buffer, pH 4.9 for 16 hr at 37°C. The enzyme reaction was stopped by incubating at 100°C for 3 min. The resulting soluble muropeptide mixture was then analyzed by ANTS labeling described below.

## In-gel fluorescence profiling of peptidoglycan fragments

*E. coli* supernatants were prepared by inoculating cultures 1:50 with an overnight culture of *E. coli* BL21-RIL(DE3) expressing SagA construct containing $His_6$ at C-terminus. Cultures were grown for 2 hr, then induced with 1 mM IPTG, and grown for an additional 2 hr. For peptidoglycan digests, 100 µg of *E. coli*, *E. faecium*, and *E. faecalis* peptidoglycan which was predigested with 20 µg mutanolysin for 16 hr at 37°C was incubated with 20 µg of purified SagA-$His_6$ overnight in PBS at 37°C. Peptidoglycan digests were dried by speed-vac before ANTS labeling. ANTS labeling was performed as described (*Jackson, 1990*). 10 µl of ANTS reaction mix was added to each tube of dried material (1:1 mixture of 0.2 M ANTS (in 3:17 acetic acid:water): 1 M $NaCNBH_3$ (in DMSO)). Reactions were incubated overnight at 37°C. 0.5–3.5 µL of the ANTS labeled mixtures were mixed 1:1 with 50% glycerol and samples were separated by native PAGE on a hand-cast 37–40% Tris-glycine acrylamide gel (19:1 polyacrylamide:bisacrylamide, with a 20% acrylamide stack) at 100 V for ~4 hr. ANTS-labeled synthetic fragments MDP, GlcNAc, MurNAc, and MurNAc-L-Ala were run for comparison. A sugar-less pentapeptide Ala-D-γ-Glu-Lys-D-Ala-D-Ala (Sigma) was run to show specificity of the UV signal and empty lanes adjacent to sample lanes were loaded with samples of ANTS labeled Ala-D-γ-Glu-Lys-D-Ala-D-Ala to prevent lane warping. Remaining lanes were loaded with 20% glycerol. Gels were imaged on the ChemiDoc MP system (Bio-Rad) using the Sybr-safe UV imaging setting.

## LC-MS analysis

For determination of the reaction products following SagA-NlpC/p60 digestion of mutanolysin-digested peptidoglycan, LC-MS analysis was carried out by the Rockefeller Proteomics Resources Center. For analysis of sample, 15 µl of digests were separated on a Acclaim 120 $C_{18}$ column (2.1 µm, 2.1 × 150 mm) (Thermo Scientific$^T$) operating at 52°C. Runs were performed at 0.2 ml/min in a mobile phase (A) of 0.1% TFA in water and an eluent (B) of 0.1% TFA in methanol using linear gradients of 0–30% B over 60 min. Products were then analyzed with an Agilent 1200 series LC/MSD TOF using electrospray ionization in positive mode, acquiring the mass range of 50 to 2000 *m/z*.

## In vitro assay of SagA-NlpC/p60 activity using HPLC-MS

Hydrolase assays were conducted in 200 µl of SagA buffer containing pure muropeptide. SagA was added at 5 µM and incubated at 37°C for 1 hr, unless indicated. The endopeptidase activity of SagA was determined with the crosslinked disaccharide tri-tetrapeptide and disaccharide tetrapeptide as substrates. All enzymatic reactions were quenched by boiling the samples for 5 min at 95°C and centrifuged at 16,000 g for 10 min to discard precipitated protein. The supernatants (reaction products) were injected into the HPLC whereas the remaining insoluble pellet was digested with muramidase and further processed for HPLC analysis as described above. Enzymatic activities were estimated from the variation in the abundance of presumed substrate and product muropeptides relative to an undigested control sample. Abundance of individual PG chains was calculated by integrating the area under the curve for each chain with the sum of all detectable peaks.

To measure specific activity, 0.5 mM of the crosslinked disaccharide tri-tetrapeptide and disaccharide tetrapeptide as substrates were incubated (total volume 50 µl) with 10 µg SagA-NlpC/p60 at 37°C for 0, 0.1, 0.25, 0.5, 1, 2 hr with purified SagA-NlpC/p60. The reaction was stopped by incubation for 5 min at 95°C. The incubation was then centrifuged for 10 min at 13000 x g and the supernatant was analyzed by LC-MS as described above.

## Crystallization, X-ray data collection, structure determination and refinement

The initial crystallization conditions for SagA-NlpC/p60 were identified using commercial screen solutions (Molecular Dimensions) by the sitting-drop vapor-diffusion method at 18°C. The final optimized crystals were obtained using a precipitant solution consisting of 2 M ammonium sulfate and 0.1 M Bis-Tris, pH 5.5, again using the sitting-drop vapor-diffusion method at 18°C. The crystals were flash-cooled in liquid nitrogen using crystallization mother liquor without additional cryoprotectant. X-ray diffraction data were collected from a single crystal on beamline AMX at NSLS-II beamline (Brookhaven National Laboratory) to 2.4 Å resolution. The data were processed with the HKL2000 program suite (*Otwinowski and Minor, 1997*). Initial phase estimates and electron-density maps were obtained by molecular replacement with Phaser (*McCoy et al., 2007*) using the C-terminus of CwlT (PDB: 4FDY) as an initial search model in Phenix (*Adams et al., 2011*). Then the structure was improved by building in COOT and refinement in Phenix iteratively (*Emsley and Cowtan, 2004*). Comprehensive model validation was performed with MolProbity (*Chen et al., 2010*) with 97.4/2.6% of residues falling within the favored and allowed region of the Ramachandran plot, respectively. (data-collection and refinement statistics are summarized in *Supplementary file 5*). All molecular graphics were prepared with PyMOL (The PyMOL Molecular Graphics System, Version 2.0 Schrödinger, LLC.). Atomic coordinates and experimental structure factors have been deposited in the PDB under accession code 6B8C. The sequence conservation analysis shown in *Figure 3* was computed using the ConSurf server (*Ashkenazy et al., 2010*). In brief, a multiple sequence alignment of SagA-NlpC/p60 to its closest 150 homologues was generated using the HHMER algorithm provided by ConSurf, with conservation scores plotted in PyMOL.

## Docking study

The Grid-based Ligand Docking with Energetics (GLIDE) module of the Schrödinger suite was used to generate a list of poses with highest scores (*Friesner et al., 2004*; *Halgren et al., 2004*). The docking search space was specified by setting the ligand diameter midpoint cubic box to $15 \times 15 \times 15$ Å$^3$, which broadly covers the whole region of the SagA-NlpC/p60 protein. The key ligand binding cysteine residues identified from previous biochemistry studies on *E. faecium_/ E. faecalis* PG digested by SagA-NlpC/p60, which showed a loss of function on mutation, were used to define the ligand recognition sites. Standard parameters were applied including van der Waals (vdW) scaling of nonpolar atoms (by 0.7) to include modest 'induced fit' effect of ligand. The molecular docking of SagA-NlpC/p60 with PG ligands was done in a flexible and non-constrained manner, allowing the ligand to move freely over the entire volume of the grid box. All other settings were kept as default and docking simulations were performed in two steps which included initial validation of the standard precision (SP) docking algorithms to predict the most accurate binding of ligand with apo SagA-NlpC/p60 structure, then used molecular mechanics generalized born surface area (MM-GBSA) calculations of binding free energies of top-ranked ligand poses (*Lyne et al., 2006*).

## Activation of NOD1/2 in mammalian cells

Human embryonic kidney 293T (HEK293T) cells were obtained from the American Type Culture Collection and tested for mycoplasma contamination. HEK293T cells were cultured in Dulbecco's Modified Eagle Medium (with D-glucose, L-glutamine and sodium pyruvate) supplemented with 10% fetal bovine serum. *E. faecium* and *E. faecalis* were cultured in BHI medium. *E. faecalis*-sagA was cultured in BHI medium supplemented with 8 µg/mL chloramphenicol. HEK293T cells were seeded in 24-well plates ($1.5–2 \times 10^5$ cells/well). After 24 hr, the cells were transfected with plasmids expressing firefly luciferase, Renilla luciferase, NOD1, (or NOD2), (PRDII-4X-luc: 50 ng/well, pRL-TK: 5 ng/well, pUNO1-hNOD1 or pUNO-1-hNOD2a: 5 ng/well, lipofectamine 2000: 2.5 µL/well, total volume is 500 µL of Opti-MEM per well). After 6 hr, the transfection medium was removed and replaced with fresh Opti-MEM containing 50 µM of iE-DAP (NOD1 ligand) or 5 µM MDP (NOD2 ligand) and the solution was incubated for 16–17 hr. Some wells were added Opti-MEM containing *E. faecium*, *E. faecalis*, *E. faecalis*-sagA at MOI = 1–1000 (the bacteria were grown until $OD_{600}$ reaches ~0.6, then washed with PBS three times and then serial diluted in Opti-MEM). After 4 hr, the media containing bacteria were removed and replaced by fresh Opti-MEM media supplemented with 250 µg/mL gentamicin, and the solution was incubated for additional 12–13 hr. The cells were lysed and assayed for luciferase activity according to the manufacture's protocol (Dual-Luciferase Reporter Assay System, Promega).

## Gentamicin protection assays

For measuring colony-forming unit (CFU), the solution was removed, and the cells were washed three times with PBS. 250 µL of Triton X-100 (1% in PBS) was added to each well. The plates were shaken gently for 10 min. The solution was serially diluted 10-fold, and 5 µL were spotted onto agar plates. The plates were placed in a 37 °C-incubator for about 16 hr (for most bacteria) or 36 hr (for *E. faecalis*-sagA) before counting colonies.

## Generation of *E. faecalis*-sagA active site mutant

*E. faecalis*-sagA active site mutant (AS) was generated as previously described (*Rangan et al., 2016*), but with these mutations C443A, H494A and H506A.

## Probiotic treatment and *C. difficile* infection experiments

C57BL/6J (000664) mice were purchased from the Jackson Laboratory and maintained at the Rockefeller University animal facilities. Mice 8 weeks of age were used for experiments. Animal care and experimentation were consistent with the National Institutes of Health guidelines and approved by the Institutional Animal Care and Use Committee of the Rockefeller University. *C. difficile* infections were performed according to the previous protocol (*Pedicord et al., 2016*). Mice were gavaged with AMNV (4 mg ampicillin, 2 mg metronidazole, 4 mg neomycin, 2 mg vancomycin) antibiotic cocktail daily for 7 days. AMNV treatment was ceased 2 days before colonization with probiotics. The evening before colonization with probiotics, *L. plantarum* WCFS1 with the pAM401 plasmid containing either empty vector or a variant of SagA was grown in MRS supplemented with 8 µg/mL chloramphenicol at 37°C overnight aerobically. The next day, *L. plantarum* cultures were pelleted and resuspended in sterile phosphate-buffered saline (PBS) at a concentration of $10^9$ cfu/mL. Mice were colonized by oral gavage with 100 µL of sterile PBS or the *L. plantarum* suspensions. Mice were then infected with *C. difficile* 36 hr after probiotic treatment. Two days before infection, *C. difficile* was streaked onto BHI agar supplemented (BHIS++) with 0.1% taurocholate, 0.1% L-cysteine, 16 µg/mL cefoxitin, and 250 µg/mL cycloserine and grown overnight at 37°C in an anaerobic chamber (*Sorg and Dineen, 2009*). The following day, a single colony from the plate was inoculated into 5 ml of BHIS ++broth and incubated overnight anaerobically at 37°C. Sterile PBS (5 ml) was added, and broth culture tubes were then parafilmed and exported from the anaerobic chamber for centrifugation at 2000 g for 5 min. The tubes were returned to the anaerobic chamber, the supernatant was removed, and the bacterial pellet was resuspended in 3 ml of sterile PBS. This suspension was loaded into 1 ml syringes fitted with gavage needles and exported from the anaerobic chamber in plastic bags. *C. difficile* suspensions were immediately transported to animal facilities for oral gavage of 100 µl per mouse. Weight loss was monitored just before infection, and mice were euthanized

when they reached 80% baseline weight or when they appeared hunched or moribund, whichever occurred first. Death was not used as an end point.

## Chemical synthesis

Compounds 1 and 2 were synthesized as previous reported (*Lioux et al., 2005*).

### Compound 3

Zinc dust (4.09 g, 62.5 mmol) was added to a solution of compound one in THF/Ac$_2$O/AcOH (3.9 mL/2.6 mL/1.3 mL). The mixture was stirred at room temperature for 5.5 hr. The mixture was filtered on celite. The filtrate was concentrated by rotary evaporation. The resulting solution was purified by silica gel column chromatography (MeOH/CHCl$_3$ = 1.5:100 to 2:100) to yield compound 3 as a white solid (0.799 g, mixture with compound S-1). $^1$H NMR (600 MHz, CD$_3$OD): δ 8.07 (d, *J* = 4.5 Hz, 1H), 8.00 (d, *J* = 7.7 Hz, 2H), 7.94 (d, *J* = 7.6 Hz, 2H, from **S-1**), 7.87 (t, *J* = 9.1 Hz, 1H), 7.73 (t, *J* = 7.7 Hz, 2H, 1H from **S-1**), 7.63 (t, *J* = 7.8 Hz, 2H, from **S-1**), 7.37 (m, 15H, 4H from **S-1**), 7.32 (m, 9H, 4H from **S-1**), 5.26 (t, *J* = 9.9 Hz, 1H), 5.16 (d, *J* = 2.6 Hz, 1H), 5.08 (d, *J* = 3.3 Hz, 1H, from **S-1**), 5.03 (t, *J* = 9.7 Hz, 1H), 4.94 (d, *J* = 3.5 Hz, 1H), 4.72 (m, 4H, 2H from **S-1**), 4.62 (m, 5H, 2H from **S-1**), 4.53 (m, 4H, 2H from **S-1**), 4.45 (m, 3H, 2H from **S-1**), 4.35 (dd, *J* = 12.5, 3.8 Hz, 1H), 4.14 (dd, *J* = 10.7, 3.6 Hz, 1H), 4.03 (t, *J* = 10.0 Hz, 3H), 3.94 (d, *J* = 5.6 Hz, 1H), 3.75 (m, 7H, 4H from **S-1**), 3.66 (m, 7H, 3H from **S-1**), 3.53 (t, *J* = 9.2 Hz, 1H from **S-1**), 3.44 (d, *J* = 10.3 Hz, 1H), 2.05 (s, 3H), 2.04 (s, 3H), 2.00 (s, 3H), 1.96 (s, 3H, from **S-1**), 1.93 (s, 3H), 1.47 (d, *J* = 6.9 Hz, 3H), 1.24 (d, *J* = 7.0 Hz, 3H from **S-1**), 1.12 (d, *J* = 7.2 Hz, 3H). $^{13}$C NMR (600 MHz, CD$_3$OD): δ 175.20, 174.54 (from **S-1**), 172.20, 172.03 171.96 (from **S-1**), 170.63, 170.30, 170.00, 139.92, 139.42 (from **S-1**), 138.42, 138.30 (from **S-1**), 138.06, 137.50 (from **S-1**), 137.11, 133.94 (from **S-1**), 133.86, 129.29, 129.21 (from **S-1**), 128.17, 128.10 (from **S-1**), 128.02 (from **S-1**), 127.98 (from **S-1**), 127.82 (from **S-1**), 127.78, 127.69, 127.50, 127.47 (from **S-1**), 127.41 (from **S-1**), 127.36, 127.26 (from **S-1**), 99.19, 96.14, 95.90 (from **S-1**), 95.81, 78.39 (from **S-1**), 75.75, 75.39, 75.34, 75.31, 75.20 (from **S-1**), 74.88, 73.26, 73.12 (from **S-1**), 72.73, 72.32, 71.92, 71.49 (from **S-1**), 71.27, 70.86, 70.72, 69.63, 69.47, 69.16 (from **S-1**), 69.12 (from **S-1**), 68.96, 68.79, 68.52, 67.82, 67.50, 61.17, 58.33, 58.08 (from **S-1**), 54.93, 54.38, 54.28 (from **S-1**), 54.08, 53.99, 53.49 (from **S-1**), 51.62, 21.42 (from **S-1**), 20.99, 19.34, 19.21, 19.17, 17.62 (from **S-1**), 17.21, 16.68. ESI-MS [M+H$^+$]: *m/z* calc. for C$_{47}$H$_{59}$N$_2$O$_{18}$S$^+$: 971.3478; found: 971.3476.

### Compound 4

DBU (0.12 mL, 0.851 mmol) was added to a solution of compound **3** (0.799 g, 0.823 mmol, max.) in CH$_2$Cl$_2$ (15 mL) at 0℃. The mixture was stirred at room temperature for 45 min. 1M HCl$_{(aq)}$ (40 mL) was added to quench the reaction. The mixture was extracted with CH$_2$Cl$_2$ (2 × 25 mL) and the combined organic layers were dried with MgSO$_4$ and concentrated by rotary evaporation. The resulting solid was washed with Et$_2$O several times and dried in vacuum to yield compound **4** as a white solid (0.516 mg, 38% over three steps). Crude compound four was used in the next step without further purification. $^1$H NMR (600 MHz, CD$_3$OD): δ 7.45 (d, *J* = 7.4 Hz, 2H), 7.41 (t, *J* = 7.5 Hz, 2H), 7.33 (m, 5H), 7.29 (m, 1H), 5.32 (t, *J* = 9.8 Hz, 1H), 5.23 (d, *J* = 3.1 Hz, 1H), 4.97 (t, *J* = 9.6 Hz, 1H), 4.81 (d, *J* = 8.3 Hz, 1H), 4.68 (m, 2H), 4.61 (d, *J* = 12.2 Hz, 1H), 4.52 (d, *J* = 12.2 Hz, 1H), 4.33 (dd, *J* = 12.5, 4.0 Hz, 1H), 4.06 (t, *J* = 9.3 Hz, 1H), 3.99 (d, *J* = 12.3 Hz, 1H), 3.74 (m, 2H), 3.67 (m, 3H), 3.45 (d, *J* = 10.3 Hz, 1H), 2.01 (s, 6H), 2.01 (s, 3H), 1.97 (s, 3H), 1.90 (s, 3H), 1.47 (d, *J* = 7.0 Hz, 3H). $^{13}$C NMR (600 MHz, CD$_3$OD): δ 177.91, 172.24, 171.94, 170.76, 170.37, 169.88, 138.44, 137.63, 128.15, 127.97, 127.62, 127.53, 127.45, 127.41, 99.19, 95.66, 76.02, 75.36, 75.20, 72.73, 71.97, 71.37, 71.00, 69.53, 68.71, 67.84, 61.32, 55.21, 54.40, 21.39, 21.26, 19.24, 19.12, 17.69. ESI-MS [M+H$^+$]: *m/z* calc. for C$_{39}$H$_{51}$N$_2$O$_{16}$$^+$: 803.3233; found: 803.3288.

### Compound 5

N-hydroxysuccinimide (0.107 g, 0.929 mmol) and N-(3-dimethylaminopropyl)-N'-ethylcarbodiimide hydrochloride (0.183 g, 0.955 mmol) were added to a solution of compound **4** (0.516 g, 0.643 mmol) in DMF (12 mL). The mixture was stirred at room temperature for 16 hr. The solvent was removed under vacuum. H$_2$O (20 mL) was added. The mixture was extracted with EtOAc (25 mL) and the organic layer was dried with MgSO$_4$ and concentrated by rotary evaporation. The resulting solution was purified by silica gel column chromatography (MeOH/CHCl$_3$ = 1:100 to 2:100) to yield

compound 5 as a white solid (0.294 g, 51%). %). $^1$H NMR (600 MHz, CDCl$_3$): δ 7.53 (m, 2H), 7.47 (m, 3H), 7.36 (d, $J$ = 4.2 Hz, 4H), 7.31 (m, 1H), 6.95 (d, $J$ = 8.1 Hz, 1H), 5.02 (m, 3H), 4.89 (m, 3H), 4.69 (d, $J$ = 12 Hz, 1H), 4.55 (d, $J$ = 12.6 Hz, 1H), 4.51 (d, $J$ = 8.4 Hz, 1H), 4.43 (d, $J$ = 12.6 Hz, 1H), 4.40 (m, 1H), 4.12 (m, 1H), 4.02 (d, $J$ = 10.2 Hz, 2H), 3.78 (m, 2H), 3.67 (m, 1H), 3.59 (dd, $J$ = 10.9, 2.3 Hz, 1H), 3.47 (m, 1H), 2.91 (s, 4H), 2.06 (s, 3H), 2.05 (s, 3H), 2.02 (s, 3H), 1.92 (s, 3H), 1.78 (s, 3H), 1.56 (d, $J$ = 6.6 Hz, 3H). $^{13}$C NMR (600 MHz, CDCl$_3$): δ 171.69, 170.68, 170.58, 170.32, 169.67, 169.38, 137.42, 137.29, 129.06, 128.96, 128.35, 127.77, 99.74, 96.54, 74.22, 73.85, 72.71, 71.64, 70.12, 69.83, 68.14, 67.61, 61.72, 54.85, 52.73, 25.63, 23.06, 22.59, 20.62, 18.08. ESI-MS [M+H$^+$]: $m/z$ calc. for C$_{43}$H$_{54}$N$_3$O$_{18}$S$^+$: 900.3397; found: 900.3381.

## Compound 6

TFA (5 mL) was added to a solution of compound two in CH$_2$Cl$_2$ (5 mL). The mixture was stirred at room temperature for 2 hr. The solvent was concentrated under vacuum. The resulting oil was dissolved in DMF (3 mL). DIEA (0.3 mL, 1.72 mmol) was added to the DMF solution. The mixture was transferred to a flask containing compound **5** (0.294 g, 0.327 mmol). The mixture was stirred at room temperature for 16 hr. DIEA (0.1 mL, 0.574 mmol) was added to the solution. The mixture was stirred at room temperature for 250 min. The solvent was removed under vacuum. H$_2$O (20 mL) and 1M HCl$_{(aq)}$ (20 mL) were added. The mixture was extracted with CH$_2$Cl$_2$ (2 × 25 mL) and the combined organic layers were dried with MgSO$_4$ and concentrated by rotary evaporation. The resulting solution was purified by silica gel column chromatography (MeOH/CHCl$_3$ = 0.4/9.6) to yield compound **6** as a white solid (0.116 g, 33%). $^1$H NMR (600 MHz, CDCl$_3$): δ 7.51 (m, 2H), 7.48 (m, 1H), 7.44 (m, 2H), 7.37 (m, 6H), 7.31 (m, 3H), 7.27 (d, $J$ = 7.6 Hz, 1H), 6.88 (d, $J$ = 5.7 Hz, 1H), 6.82 (s, 1H), 5.81 (s, 1H), 5.13 (m, 3H), 5.01 (m, 2H), 4.89 (t, $J$ = 10.0 Hz, 1H), 4.85 (d, $J$ = 11.9 Hz, 1H), 4.65 (d, $J$ = 12.4 Hz, 1H), 4.53 (d, $J$ = 11.9 Hz, 1H), 4.41 (d, $J$ = 11.9 Hz, 1H), 4.37 (m, 3H), 4.28 (dd, $J$ = 12.4, 4.7 Hz, 1H), 4.24 (t, $J$ = 6.9 Hz, 1H), 4.01 (m, 2H), 3.93 (m, 2H), 3.65 (d, $J$ = 10.0 Hz, 1H), 3.58 (dd, $J$ = 10.9, 2.8 Hz, 1H), 3.46 (m, 3H), 2.55 (m, 1H), 2.37 (m, 1H), 2.21 (m, 1H), 2.06 (s, 3H), 2.05 (s, 3H), 2.02 (s, 3H), 1.99 (m, 1H), 1.94 (s, 3H), 1.80 (s, 3H), 1.42 (d, $J$ = 7.1 Hz, 3H), 1.38 (d, $J$ = 6.7 Hz, 3H). $^{13}$C NMR (600 MHz, CDCl$_3$): δ 174.76, 173.74, 173.50, 172.63, 170.93, 170.89, 170.84, 170.54, 169.40, 137.62, 137.08, 135.66, 128.94, 128.82, 128.62, 128.56, 128.37, 128.27, 128.08, 99.45, 96.68, 76.49, 75.10, 73.80, 72.60, 71.58, 70.26, 70.07, 68.40, 67.76, 66.62, 61.70, 54.45, 53.82, 52.67, 49.81, 30.73, 26.35, 23.16, 23.05, 20.72, 20.64, 20.62, 18.72, 16.92. ESI-MS [M+H$^+$]: m/z calc. for C$_{54}$H$_{70}$N$_5$O$_{19}$$^+$: 1092.4660; found: 1092.4676.

## GlcNAc-MurNAc-L-Ala-D-isoGln

10% NaOH$_{(aq)}$ (0.2 mL, 0.5 mmol) was added to a suspension compound **6** in 1,4-dioxane/MeOH/H$_2$O (1 mL/0.9 mL/0.1 mL). The mixture was stirred at room temperature for 80 min. H$_2$O (3 mL) and 1M HCl$_{(aq)}$ (0.6 mL) were added. The mixture was lyophilized for 24 hr. The resulting residue was dissolved in MeOH (3 mL). 10% palladium on carbon (22 mg) was added to the solution. The mixture was stirred under 1 atm. of H$_2$ at room temperature for 16 hr. After filtration, the solution was concentrated by rotary evaporation. The resulting solution was purified by reversed-phase C-18 HPLC (gradient 0–15%, 0.1% TFA in H$_2$O/CH$_3$CN, over 50 min) to yield compound GlcNAc-MurNAc-L-Ala-D-isoGln as a white solid (14.6 mg, 20%). $^1$H NMR (600 MHz, CD$_3$OD): δ 5.27 (d, $J$ = 3 Hz, 1H-α), 4.66 (d, $J$ = 6.6 Hz, 1H-β), 4.60–4.45 (m, 3H), 4.37 (d, $J$ = 7.2 Hz, 1H), 3.92 (m, 2H), 3.78 (m, 3H), 3.74 (m, 3H), 3.50 (m, 1H), 3.25 (m, 1H), 2.45–2.20 (m, 4H) 2.02 (s, 3H), 1.98 (s, 3H), 1.45 (d, $J$ = 7.2 Hz, 1H), 1.43 (d, $J$ = 6.6 Hz, 3H). $^{13}$C NMR (600 MHz, CD$_3$OD): δ 176.27, 175.25, 173.72, 173.10, 172.53, 172.25, 100.59, 96.23, 90.54, 78.36, 76.89, 76.68, 75.84, 75.53, 75.01, 74.21, 71.31, 61.76, 60.12, 56.36, 54.35, 51.77, 49.50, 31.22, 27.03, 21.63, 21.47, 18.14, 16.52. ESI-MS [M+H$^+$]: $m/z$ calc. for C$_{27}$H$_{46}$N$_5$O$_{16}$$^+$: 696.2934; found: 696.2918.

## Acknowledgements

The authors thank Joseph P Fernandez, Brian D Dill and Henrik Molina for assistance with peptidoglycan analysis (The Rockefeller University Proteomics Resource Center). The authors also thank Uhn-Soo Cho (University of Michigan, Ann Arbor), Yehuda Goldgur (MSKCC) for assistance in SagA-NlpC/p60 structural determination. Nadine Soplop and Kunihiro Uryu (Electron Microscopy Resource

Center at The Rockefeller University) helped for electron microscopy studies. We thank Hang lab members for helpful comments on the manuscript. We thank Daniel Mucida (Rockefeller University) for supporting this study and feedback on manuscript. BK was supported by Helmsley postdoctoral fellowship. YCW was a Cancer Research Institute Irvington Fellow supported by Cancer Research Institute. CWH acknowledges support from NIH-NICCH F32 AT010087-01A1 grant. JE and KR were supported by The Rockefeller University Graduate Program. The project described was also supported in part by grant UL1 TR0018663 from the National Center for Advancing Translational Sciences (NCATS, National Institutes of Health (NIH) Clinical and Translational Science Award (CTSA) program. It was supported by Award Number S10RR031855 from the National Center for Research Resources. The use of the formulator and the phoenix robots in the Rockefeller University Structural Biology Resource Center was made possible by Grant Number 1S10RR027037-01 from the National Center for Research Resources of the NIH. The beam line AMX of the NSLS-II by Brookhaven National Laboratory was supported by the US Department of Energy under Contract No. DE-SC0012704. HCH acknowledges support from NIH-NIGMS R01 GM103593 grant and Robertson Therapeutic Fund.

## Additional information

### Funding

| Funder | Grant reference number | Author |
|---|---|---|
| Leona M. and Harry B. Helmsley Charitable Trust | Postdoctoral fellowship | Byungchul Kim |
| National Center for Complementary and Integrative Health | F32AT010087 | Charles W Hespen |
| National Institute of General Medical Sciences | R01GM103593 | Howard C Hang |
| Cancer Research Institute | Postdoctoral fellowship | Yen-Chih Wang |

The funders had no role in study design, data collection and interpretation, or the decision to submit the work for publication.

### Author contributions

Byungchul Kim, Conceptualization, Data curation, Formal analysis, Investigation, Methodology, Writing—original draft, Writing—review and editing; Yen-Chih Wang, Charles W Hespen, Conceptualization, Data curation, Formal analysis, Investigation, Methodology, Writing—review and editing; Juliel Espinosa, Jeanne Salje, Kavita J Rangan, Deena A Oren, Data curation, Formal analysis, Investigation, Methodology, Writing—review and editing; Jin Young Kang, Formal analysis, Validation, Investigation, Methodology, Writing—review and editing; Virginia A Pedicord, Formal analysis, Investigation, Methodology, Writing—review and editing; Howard C Hang, Conceptualization, Data curation, Formal analysis, Supervision, Funding acquisition, Writing—original draft, Project administration, Writing—review and editing

### Author ORCIDs

Byungchul Kim http://orcid.org/0000-0002-0207-7357
Charles W Hespen http://orcid.org/0000-0001-5466-0280
Juliel Espinosa http://orcid.org/0000-0003-2104-5331
Howard C Hang http://orcid.org/0000-0003-4053-5547

### Decision letter and Author response

Decision letter https://doi.org/10.7554/eLife.45343.038
Author response https://doi.org/10.7554/eLife.45343.039

## Additional files

#### Supplementary files

• Supplementary file 1. Molecular mass and composition of muropeptides from *E. faecium*. (**a**) Peak numbers refer to *Figure 2*-b. (**b**) GM, disaccharide (GlcNAc-MurNAc); 2 GM, disaccharide-disaccharide (GlcNAc-MurNAc-GlcNAc-MurNAc); 3 GM, disaccharide-disaccharide-disaccharide (GlcNAc-MurNAc-GlcNAc-MurNAc- GlcNAc-MurNAc); GM-Tri, disaccharide tripeptide (L-Ala-D-iGln-L-Lys); GM-Tetra, disaccharide tetrapeptide (L-Ala-D-iGln-L-Lys-D-Ala); GM-Penta, disaccharide pentapeptide (L-Ala-D-iGln-L-Lys-D-Ala -D-Ala).

DOI: https://doi.org/10.7554/eLife.45343.027

• Supplementary file 2. Molecular mass and composition of muropeptides from *E. faecalis* and *E. faecalis-sagA*. (**a**) Peak numbers refer to *Figure 2*-b. (**b**) GM, disaccharide (GlcNAc-MurNAc); 2 GM, disaccharide-disaccharide (GlcNAc-MurNAc-GlcNAc-MurNAc); 3 GM, disaccharide-disaccharide-disaccharide (GlcNAc-MurNAc-GlcNAc-MurNAc- GlcNAc-MurNAc); GM-Tri, disaccharide tripeptide (L-Ala-D-iGln-L-Lys); GM-Tetra, disaccharide tetrapeptide (L-Ala-D-iGln-L-Lys-D-Ala); GM-Penta, disaccharide pentapeptide (L-Ala-D-iGln-L-Lys-D-Ala -D-Ala). (**c**) ND: Precise structure unknown. d. The assignment of the amide and the hydroxyl functions to either peptide stem is arbitrary.

DOI: https://doi.org/10.7554/eLife.45343.028

• Supplementary file 3. List of genes that have synonymous and non-synonymous mutations in *E. faecalis-sagA* compared with *E. faecalis*. All sequencing data are available from GenBank under accession number CP025022, CP025020, and CP025021 for *Enterococcus faecium* Com15, *Enterococcus faecalis* OG1RF, and *Enterococcus faecalis* OG1RF-*sagA*.

DOI: https://doi.org/10.7554/eLife.45343.029

• Supplementary file 4. Molecular mass and composition of enzymatic products from incubation of *E. faecalis* PG with purified SagA-NlpC/p60 domain. (**a**) Peak numbers refer to *Supplementary file 5e*. (**b**) GM, disaccharide (GlcNAc-MurNAc); GM-di, disaccharide dipeptide (L-Ala-D-iGln); GM-Tri, disaccharide tripeptide (L-Ala-D-iGln-L-Lys); GM-Tetra, disaccharide tetrapeptide (L-Ala-D-iGln-L-Lys-D-Ala).

DOI: https://doi.org/10.7554/eLife.45343.030

• Supplementary file 5. Crystallographic statistics. (**a**) One crystal was used to determine structure. (**b**) Values in parentheses are for highest resolution shell.

DOI: https://doi.org/10.7554/eLife.45343.031

• Supplementary file 6. Structural comparisons of SagA-NlpC/p60 domain with structurally similar homologs determined by DALI server. The structural alignment was performed by the DALI server (*Holm and Sander, 1995*). For structures with multiple chains/models, only results for the first structure with the highest Z-score are shown. No. of residues: the number of residues present in the model used for comparison; Seq id: percentage sequence identity of the pairwise structural alignment.

DOI: https://doi.org/10.7554/eLife.45343.032

• Supplementary file 7. Predicted binding free energies of highest-scoring poses of docked GlcNAc-MurNAc-L-Ala-D-isoGln-L-Lys-D-Ala as generated with MM-GBSA. MM-GBSA calculations were carried out using the Prime_MM-GBSA utility.

DOI: https://doi.org/10.7554/eLife.45343.033

• Transparent reporting form

DOI: https://doi.org/10.7554/eLife.45343.034

#### Data availability

All data generated or analyzed during this study are included in the manuscript and supporting files. Diffraction data have been deposited in PDB under the accession code 6B8C.

The following dataset was generated:

| Author(s) | Year | Dataset title | Dataset URL | Database and Identifier |
|---|---|---|---|---|
| Kim B, Oren D | 2019 | Crystal structure of NlpC/p60 domain of peptidoglycan hydrolase | http://www.rcsb.org/structure/6B8C | Protein Data Bank, 6B8C |

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
