## [Decision Letter]

Thank you for submitting your article "*Enterococcus faecium* secreted antigen A generates muropeptides that activate NOD2-dependent immune signaling" for consideration by *eLife*. Your article has been reviewed by three peer reviewers, including Kim Orth as the Reviewing Editor and Reviewer #1, and the evaluation has been overseen by Wendy Garrett as the Senior Editor. The following individuals involved in review of your submission have agreed to reveal their identity: Daria Van Tyne (Reviewer #2); Nicole M Koropatkin (Reviewer #3).

Summary:

General assessment and major comments. The study by Kim et al. seeks to elucidate the mechanism by which the *E. faecium* peptidoglycan hydrolase SagA enhances intestinal barrier function and pathogen tolerance in *C. elegans* and in mice. Analysis of data from muropeptide fragment analysis, x-ray crystallography, and NOD1 and NOD2 activation assays leads the authors to conclude that SagA hydrolyzes peptidoglycan into small muropeptide fragments, which activate NOD2 with high efficiency. Overall the paper represents a very significant advance, is of broad interest, and flows well logically. Below are a list of revisions requested by the three reviewers.

Essential revisions:

1) The authors compared their structure to homologs in the PDB but no references were provided for these (maybe they are to be published). Please address. Also not much context for the substrates of these other enzymes was provided so please provide additional rationale. From a structural biology perspective, a little more depth in the comparison (maybe limiting to one or two more meaningful comparisons) would provide better context as the glycan binding site of the SagA-NlpC/p60 domain is not conserved.

2) The authors should provide representative electron density for some part of their structure – perhaps the active site to demonstrate the quality of the data. The statistics are quite good though the Rw/Rf were a little elevated.

3) Which "pose" is being displayed for the modeled ligand (Supplementary file 7)? From the renderings of the model, it was difficult to discern how well residues within the proposed binding pocket accommodate the ligand as only general shape recognition (which looks good) is displayed.

4) Please expand on the statement in the Discussion that the N-terminal coil-coil of homologous enzymes have been implicated in autoinhibition.

5) The authors present convincing data supporting the proposed molecular mechanism for SagA activity, but the link between this activity and enhanced intestinal barrier function or pathogen tolerance is merely implied. Given that this research group has previously used a straightforward *C. elegans* assay to measure these phenotypes, the authors should test whether non-crosslinked muropeptides alone can enhance intestinal barrier function or improve pathogen tolerance in worms. This would show that the proposed mechanism of action of SagA is truly relevant in vivo.

6) In Figure 1B, were replicate experiments conducted? If not they should be, and error bars should be plotted for each OD value. Also consider applying an appropriate statistical test to show that the difference in growth rate between Efl and Efl-sagA is significant.

7) Many more *E. faecalis* genomes are now available compared to those that were published by Palmer et al. in 2010. The absence of SagA in all or most *E. faecalis* strains should be confirmed through a standard BLAST of the NCBI *E. faecalis* WGS database, which would be a more appropriate analysis.

8) Based on these findings, it is somewhat perplexing why SagA is an essential gene. The authors should revisit making a clean deletion of SagA to check the growth phenotype.

---

## [Author Response]

Essential revisions:1) The authors compared their structure to homologs in the PDB but no references were provided for these (maybe they are to be published). Please address. Also not much context for the substrates of these other enzymes was provided so please provide additional rationale. From a structural biology perspective, a little more depth in the comparison (maybe limiting to one or two more meaningful comparisons) would provide better context as the glycan binding site of the SagA-NlpC/p60 domain is not conserved.

We thank the reviewers for these suggestions and have now added the references to other NlpC/p60 hydrolases (which should have been included originally) and included a direct comparison of the putative substrate binding site of SagA-NlpC/p60 with CwlT in new Figure 4A as well as additional text in manuscript.

2) The authors should provide representative electron density for some part of their structure – perhaps the active site to demonstrate the quality of the data. The statistics are quite good though the Rw/Rf were a little elevated.

We thank the reviewers for this suggestion and have now included zoomed in electron density of the active site catalytic triad in new Figure 3E.

3) Which "pose" is being displayed for the modeled ligand (Supplementary file 7)? From the renderings of the model, it was difficult to discern how well residues within the proposed binding pocket accommodate the ligand as only general shape recognition (which looks good) is displayed.

We thank the reviewers for this question and have noted that pose 1 from Supplementary file 7 is shown in new Figure 4C.

4) Please expand on the statement in the Discussion that the N-terminal coil-coil of homologous enzymes have been implicated in autoinhibition.

We thank the reviewers for this suggestion and have now added additional discussion on N-terminal coil-coil of SagA.

*5) The authors present convincing data supporting the proposed molecular mechanism for SagA activity, but the link between this activity and enhanced intestinal barrier function or pathogen tolerance is merely implied. Given that this research group has previously used a straightforward C. elegans assay to measure these phenotypes, the authors should test whether non-crosslinked muropeptides alone can enhance intestinal barrier function or improve pathogen tolerance in worms. This would show that the proposed mechanism of action of SagA is truly relevant* in vivo.

With regard to the activity of crosslinked and non-crosslinked muropeptides in vivo, although worms do not have NOD2 ortholog we previously demonstrated MurNAc-L-Ala, MurNAc, but not MDP or GlcNAc, were protective in *C. elegans* (Rangan et al., 2016) and have added this to the Discussion. We are actually exploring the direct MurNAc receptor in worms with photoaffinity analogs, but those studies are beyond the scope of this manuscript.

To provide in vivo evidence that SagA secretion and activity are important in vivo, we have added recent data from the lab in new Figures 5C and D, which shows that signal sequence and active site mutants of SagA can be expressed in the probiotic L. plantarum but are not protective against C. difficile infection in mice, while wild-type SagA is protective. This provides additional support for our model of SagA activity with peptidoglycan fragments in vivo, shown in new Figure 6.

6) In Figure 1B, were replicate experiments conducted? If not they should be, and error bars should be plotted for each OD value. Also consider applying an appropriate statistical test to show that the difference in growth rate between Efl and Efl-sagA is significant.

We thank the reviewers for this suggestion and have now added the error and statistics to the data in Figure 1B.

7) Many more E. faecalis genomes are now available compared to those that were published by Palmer et al. in 2010. The absence of SagA in all or most *E. faecalis* strains should be confirmed through a standard BLAST of the NCBI *E. faecalis* WGS database, which would be a more appropriate analysis.

We thank the reviewers for this suggestion and we have now evaluated additional *E. faecalis* genomes and putative NlpC/p60 that are well annotated. We have also focused on the analysis of SagA orthologs in *E. faecium* strains, since the activity of the other SagA orthologs in other Enterococci are still unclear and in progress in our laboratory. We have replaced these supplementary figures with more focused comparisons in Figure 3—figure supplement 1 and 2.

8) Based on these findings, it is somewhat perplexing why SagA is an essential gene. The authors should revisit making a clean deletion of SagA to check the growth phenotype.

We have actually tried to remake the deletion of SagA in Efm (Com15) but have not been successful in obtaining viable bacteria, which is noted in the Discussion. We are exploring conditional approaches with CRISPR, but this involves the development of new reagents and methods that is also beyond the scope of this manuscript.

We generated Efs-SagA-active site mutant, but it was expressed at much lower levels and was thus not sufficient for functional studies, which in now included as new Figure 5—figure supplement 2 and noted in the text.